# Semantically Consistent Video Inpainting with Conditional Diffusion Models

## Abstract

Current state-of-the-art methods for video inpainting typically rely on optical flow or attention-based approaches to inpaint masked regions by propagating visual information across frames. While such approaches have led to significant progress on standard benchmarks, they struggle with tasks that require the synthesis of novel content that is not present in other frames. In this paper, we reframe video inpainting as a conditional generative modeling problem and present a framework for solving such problems with conditional video diffusion models. We introduce inpainting-specific sampling schemes which capture crucial long-range dependencies in the context, and devise a novel method for conditioning on the known pixels in incomplete frames. We highlight the advantages of using a generative approach for this task, showing that our method is capable of generating diverse, high-quality inpaintings and synthesizing new content that is spatially, temporally, and semantically consistent with the provided context.

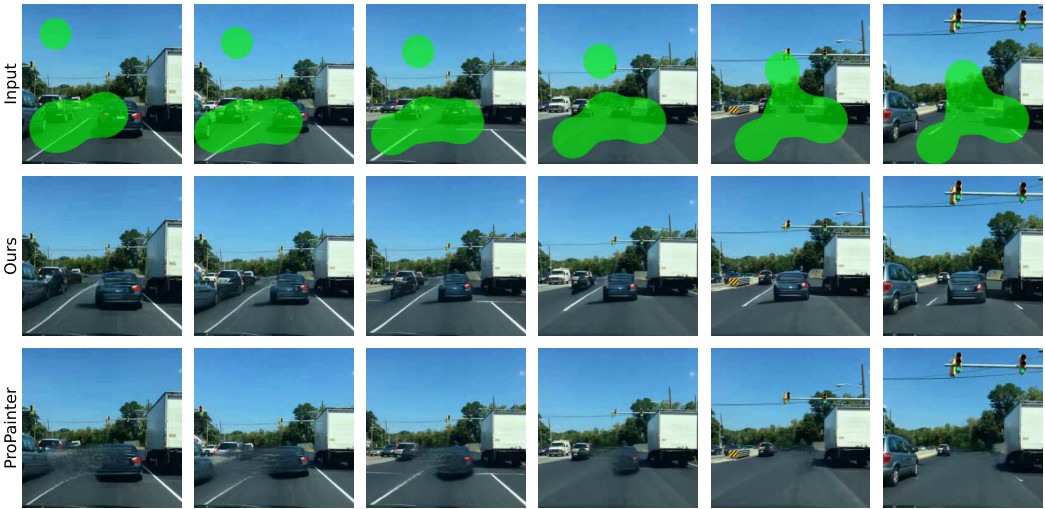

Figure 1: Inpainting results on a challenging example from our BDD-Inpainting dataset. The first row shows the input to the model, with the occlusion mask shown in green. Note that the left side of the car in the center lane is never visible, and it becomes fully occluded soon after the first frame. Our method (second row) is capable of generating a plausible completion of the car and realistically propagating it through time. On the contrary, in the result from the best-competing method, ProPainter (Zhou et al., 2023) (third row), the car is not completed and quickly fades away.

## 1 INTRODUCTION

Video inpainting is the task of filling in missing pixels in a video with plausible values. It has practical applications in video editing and visual effects, including video restoration (Tang et al., 2011), object or watermark removal (Patwardhan et al., 2005), and video stabilization (Matsushita et al., 2006). While significant progress has been made on image inpainting in recent years (Saharia et al., 2022; Lugmayr et al., 2022; Suvorov et al., 2021), video inpainting remains a challenging task due to the added time dimension, which drastically increases computational complexity and leads to a stricter notion of what it means for an inpainting to be plausible. Specifically, inpainted regions require not only per-frame spatial and semantic consistency with the context as in image inpainting, but also temporal consistency between frames and realistic motion of objects in the scene.

Current state-of-the-art video inpainting methods explicitly attempt to inpaint masked regions by exploiting visual information present in other frames, typically by using optical flow estimates (Li et al., 2022; Gao et al., 2020) or attention-based approaches (Zeng et al., 2020; Liu et al., 2021) to determine how this information should be propagated across frames. Such methods, which we refer to as "content propagation" methods, implicitly assume that the content in unmasked regions is sufficient to fill in the missing values, and tend to fail when this assumption is violated. In the presence of large occlusions, objects may be partially or fully occluded for long durations, and in these cases an inpainting method may need to complete an object's appearance (as in Fig. 1) or infer its behaviour (as in Fig. 2) to produce a plausible result, requiring the generation of novel content.

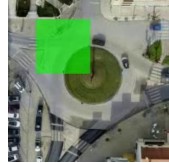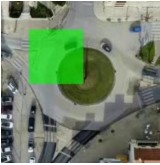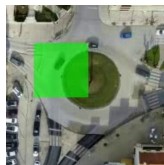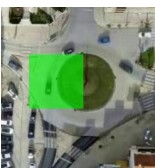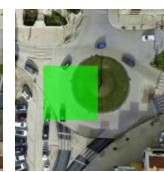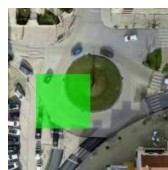

Figure 2: An example inpainting task from our Traffic-Scenes dataset. The black car becomes occluded near the top of the roundabout and emerges many frames later near the bottom. Inpainting such an example requires the ability to model plausible vehicle behaviour.

We argue that a more sensible approach to video inpainting is to learn a conditional distribution over possible inpaintings given the observed context. Conditional generative models such as GANs (Suvorov et al., 2021), autoregressive models (Peng et al., 2021), and diffusion models (Saharia et al., 2022; Lugmayr et al., 2022) have long dominated image inpainting, a task that inherently requires content synthesis as there is no additional information to draw upon. Learning such a distribution naturally accounts ill-posedness of the video inpainting problem: for instance, given the observed entry and exit points of the car in Fig. 2 there exists a diversity of plausible trajectories it could take. Further, generative approaches for image inpainting have a demonstrated ability to produce results that are *semantically* consistent with the context, a quality which is particularly important for video inpainting tasks that require inferring an object's behaviour as in Fig. 2 – without a strong prior, there is insufficient information in the context to determine what such a trajectory might look like; the model must have some notion of what makes a vehicle trajectory plausible at a semantic level.

In this work we aim to overcome the inherent limitations of content propagation by advancing generative inpainting, making the following key contributions:

- We reframe video inpainting as a conditional generative modeling problem, and present a framework for solving this problem using conditional video diffusion models.

- We demonstrate how to use long-range temporal attention to generate semantically consistent behaviour of inpainted objects over long time horizons, even when our model cannot jointly model all frames due to memory constraints. We can do this even for inpainted objects that have limited or no visibility in the context, a quality not present in the current literature.

- We introduce inpainting-specific sampling schemes which capture crucial long-range dependencies in the context, and devise a novel method for conditioning on the known pixels in incomplete frames.

- We report strong experimental results on several challenging video inpainting tasks, outperforming state-of-the-art approaches on a range of standard metrics.

- We release the BDD-Inpainting dataset, a large scale video inpainting dataset with two test sets representing a challenging new benchmark for video inpainting methods.

## 2 RELATED WORK

**Video Inpainting.** Recent advances in video inpainting have largely been driven by methods which fill in masked regions by borrowing content from the unmasked regions of other frames, which we refer to as content propagation methods. These methods typically use optical flow estimates (Huang et al., 2016; Kim et al., 2019; Xu et al., 2019; Gao et al., 2020), self-attention (Zeng et al., 2020; Liu et al., 2021; Oh et al., 2019; Lee et al., 2019) or a combination of both (Zhou et al., 2023; Zhang et al., 2022; Li et al., 2022) to determine how to propagate pixel values or learned features across frames. Such methods often produce visually compelling results, particularly on tasks where the mask-occluded region is visible in nearby frames such as foreground object removal with a near-static background. They struggle, however, in cases where this does not hold, for instance in the presence of heavy camera motion, large masks, or tasks where semantic understanding of the video content is required to produce a convincing result.

More recent work has utilized diffusion models for video inpainting. Gu et al. (2023) proposes a method for video inpainting that combines a video diffusion model with optical flow guidance, following a similar "content propagation" approach to the methods listed above. Chang et al. (2023) uses a latent diffusion model (Rombach et al., 2022; Vahdat et al., 2021) to remove the agent's view of itself from egocentric videos for applications in robotics. Notably, this is framed as an image inpainting task, where the goal is to remove the agent (with a mask provided by a segmentation model) from a single video frame conditioned on $h$ previous frames. Consequently, the results lack temporal consistency when viewed as videos, and the model is evaluated using image inpainting metrics only. Zhang et al. (2023b) proposes a method for the related task of text-conditional video inpainting, which produces impressive results but requires user intervention.

**Image Inpainting with Diffusion Models.** This work takes inspiration from the recent success of diffusion models for image inpainting. These methods can be split into two groups: those that inpaint using an unconditional diffusion model by making heuristic adjustments to the sampling procedure (Lugmayr et al., 2022; Zhang et al., 2023a), and those that explicitly train a conditional diffusion model which, if sufficiently expressive and trained to optimality, enables exact sampling from the conditional distribution (Saharia et al., 2022). In this work we extend the latter approach to video data.

## 3 BACKGROUND

**Conditional Diffusion Models.** A conditional diffusion model (Tashiro et al., 2021; Ho et al., 2020; Sohl-Dickstein et al., 2015) is a generative model parameterized by a neural network trained to remove noise from data. The network is conditioned on $t \in \{1, \ldots, T\}$, an integer describing how much noise has been added to the data. Given hyperparameters $1 > \bar{\alpha}_1 > \ldots > \bar{\alpha}_T > 0$, training data is created by multiplying the original data by a factor $\sqrt{\bar{\alpha}_t}$ and then adding unit Gaussian noise $\epsilon$ scaled to have variance $(1 - \bar{\alpha}_t^2)$. The network should then map from this noisy data, the timestep $t$, and conditioning input $\mathbf{y}$, to a prediction of the added noise $\epsilon$. It is trained with the squared error loss

$$\mathcal{L}_{\mathrm{CDM}}(\theta) := \mathbb{E}_{t,\mathbf{x},\mathbf{y},\epsilon}\left[\left\|\epsilon - \epsilon_\theta\left(\sqrt{\bar{\alpha}_t}\mathbf{x} + \sqrt{1 - \bar{\alpha}_t}\epsilon, \mathbf{y}, t\right)\right\|^2\right], \quad (1)$$

where $\epsilon_\theta(\ldots)$ is the output of the network. Data $\mathbf{x}$ and $\mathbf{y}$ are sampled from the data distribution $p_{\mathrm{data}}$, and the timestep $t$ is typically sampled from a pre-specified categorical distribution. Once such a network has been trained, various methods exist for using it to draw approximate samples from $p_{\mathrm{data}}(\mathbf{x}|\mathbf{y})$ (Ho et al., 2020; Sohl-Dickstein et al., 2015; Tashiro et al., 2021; Song et al., 2020; Karras et al., 2022).

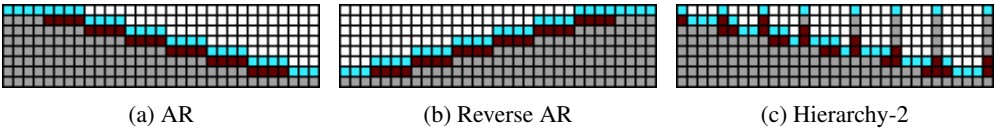

|(a) AR|(b) Reverse AR|(c) Hierarchy-2|

Figure 3: Sampling schemes from Harvey et al. (2022) for generating videos of length $N = 31$ while accessing only $K = 8$ frames at a time. Each row of each subfigure depicts a different stage of our sampling process, starting from the top row and working down. Each column represents one video frame. Within each stage, frames shown in cyan are being sampled conditioned on the values of previously-sampled frames shown in dark red. Frames shown in white are not yet generated. By the end, all frames are generated and shown in light gray.

**Video Diffusion Models.** A number of recent papers have proposed diffusion-based approaches to generative modeling of video data (Höppe et al., 2022; Harvey et al., 2022; Ho et al., 2022; Yang et al., 2022; Voleti et al., 2022). We follow the majority of these approaches in using a 4-D U-Net (Ronneberger et al., 2015) architecture to parameterize $\epsilon_\theta(\ldots)$. Alternating spatial and temporal attention blocks within the U-Net capture dependencies within and across frames respectively, with relative positional encodings (Shaw et al., 2018; Wu et al., 2021) providing information about each frame's position within the video. Due to computational constraints, video diffusion models are inherently limited to conditioning on and generating a small number of frames at a time, which we denote as $K$.

**Flexible Video Diffusion Models.** Generating long videos with frame count $N \gg K$ then requires sampling from the diffusion model multiple times. A typical approach is to generate frames in multiple stages in a block-autoregressive fashion, in each stage sampling $K/2$ frames conditioned on the previous $K/2$ frames. We depict this approach in Fig. 3a, with each row representing one stage. A problem with this strategy is that it fails to capture dependencies on frames more than $K/2$ frames in the past. Alternative orders in which to generate frames (which we refer to as "sampling schemes") are possible, as depicted in Figs. 3b and 3c. To avoid iteratively retraining to identify an optimal sampling scheme, Harvey et al. (2022) proposes training a single model that can perform well using any sampling scheme, referred to as a Flexible Diffusion Model (FDM). An FDM is trained to generate any subset of video frames conditioned on any other subset with the objective

$$\mathcal{L}_{\text{FDM}}(\theta) := \mathbb{E}_{t,\mathbf{x},\mathbf{y},\mathcal{X},\mathcal{Y},\epsilon} \left[ \left\| \epsilon - \epsilon_\theta \left( \sqrt{\bar{\alpha}_t}\mathbf{x} + \sqrt{1 - \bar{\alpha}_t}\epsilon, \mathbf{y}, \mathcal{X}, \mathcal{Y}, t \right) \right\|^2 \right], \tag{2}$$

in which $\mathbf{x}$ are the frames to remove noise from at indices $\mathcal{X}$, and $\mathbf{y}$ are the frames to condition on at indices $\mathcal{Y}$. Sampling $\mathbf{x}$ and $\mathbf{y}$ is accomplished by first sampling $\mathcal{X}$ and $\mathcal{Y}$, then sampling a training video, and then extracting the frames at indices $\mathcal{X}$ and $\mathcal{Y}$ to form $\mathbf{x}$ and $\mathbf{y}$, respectively. Once a network is trained to make predictions for arbitrary indices $\mathcal{X}$ given arbitrary indices $\mathcal{Y}$, it can be used to sample videos with any desired sampling scheme. Sampling schemes will henceforth be denoted as $\{\mathcal{X}_s, \mathcal{Y}_s\}_{s=1}^S$ where $S$ is the number of sampling stages and, at stage $s$, $\mathcal{X}_s$ and $\mathcal{Y}_s$ are the indices of the latent and observed frames, respectively.

## 4 VIDEO INPAINTING WITH CONDITIONAL DIFFUSION MODELS

### 4.1 PROBLEM FORMULATION

We consider the problem of creating an $N$-frame video $\mathbf{V}$ conditioned on some subset of known pixel values specified by a pixel-level mask $\mathbf{M}$. Entries of $\mathbf{M}$ take value 1 where the corresponding pixel in $\mathbf{V}$ is known and take 0 elsewhere. Unlike content propagation methods which deterministically generate one possible completion, we aim to model the posterior distribution over possible inpaintings as $p_\theta(\mathbf{V}|\mathbf{V} \odot \mathbf{M}) \approx p_{\text{data}}(\mathbf{V}|\mathbf{V} \odot \mathbf{M})$ with a conditional video diffusion model, where $\odot$ is defined such that $a \odot \mathbf{M}$ returns the elements in $a$ for which the corresponding value in $\mathbf{M}$ is 1.

Recall that, due to memory constraints, video diffusion models are typically restricted to conditioning on or generating at most $K$ frames at a time. In the video inpainting problem we are predicting and conditioning on pixels rather than frames, but our network architecture imposes an analogous constraint: we can only predict or condition on pixels from at most $K$ different frames at a time. We modify the definition of a sampling scheme from FDM (Harvey et al., 2022) as follows: we again

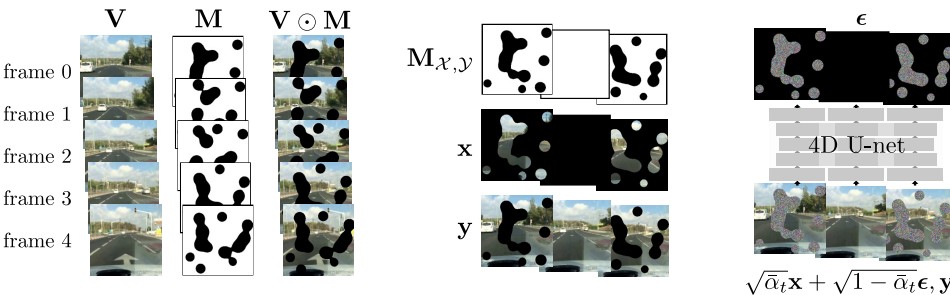

Figure 4: Example model inputs during training. **Left:** Visualizations of a 5-frame video $\mathbf{V}$, a corresponding mask $\mathbf{M}$, and the resulting known pixel values $\mathbf{V} \odot \mathbf{M}$. **Center:** Collated training inputs if $\mathcal{X} = [0, 3]$ and $\mathcal{Y} = [2]$. The observations in $\mathbf{y}$ are then the whole of frame 2 and known pixel values in frames 0 and 3. The task is to predict the unknown pixel values in frames 0 and 3. **Right:** Inputs fed to the neural network, with noise added to pixel values in $\mathbf{x}$ but not those in $\mathbf{y}$. The task is then to predict the noise $\epsilon$. For simplicity we do not show inputs $t$, $\mathbf{M}_{\mathcal{X},\mathcal{Y}}$, or $\mathcal{X} \oplus \mathcal{Y}$.

denote sampling schemes $\{\mathcal{X}_s, \mathcal{Y}_s\}_{s=1}^{S}$ with $\mathcal{X}_s$ and $\mathcal{Y}_s$ being collections of frame indices. Now, however, at each stage we sample values for only unknown pixels in frames indexed by $\mathcal{X}_s$, and condition on known pixel values in all frames indexed by either $\mathcal{X}_s$ or $\mathcal{Y}_s$. Referencing Fig. 6, in each row (stage) we show frames indexed by $\mathcal{X}_s$ in cyan and frames indexed by $\mathcal{Y}_s$ in either dark red or bright red. Frames shown in bright red contain missing pixels, which we do not wish to sample or condition on until a later sampling stage; we describe how we deal with this in Section 4.3.

## 4.2 METHOD

**Architecture.** We generalize the FDM architecture to use pixel-level masks rather than frame-level masks, as in the image inpainting approach proposed by Saharia et al. (2022). Concretely, every input frame is concatenated with a mask which takes value 1 where the corresponding pixel is observed and 0 elsewhere. The input values are clean (without added noise) for observed pixels and noisy otherwise.

**Model Inputs.** Recall that we wish to train a model that can generate plausible values for unknown pixels in frames indexed by $\mathcal{X}$, conditioned on known pixel values in frames indexed by either $\mathcal{X}$ or $\mathcal{Y}$. We simulate such tasks by first sampling a video $\mathbf{V}$ and a mask $\mathbf{M}$ from our dataset, and then sampling frame indices $\mathcal{X}$ and $\mathcal{Y}$ from a "frame index distribution" similar to that of FDM.[1] The distribution over masks $\mathbf{M}$ can be understood as reflecting the variability in the types of masks we will encounter at test-time, and the frame index distribution can be understood as reflecting our desire to be able to sample from the model using arbitrary sampling schemes. Given $\mathbf{M}$, $\mathcal{X}$, and $\mathcal{Y}$, we create a combined list of frames $\mathcal{X} \oplus \mathcal{Y}$, where $\oplus$ denotes concatenation, and a corresponding mask $\mathbf{M}_{\mathcal{X},\mathcal{Y}} := \mathbf{M}[\mathcal{X}] \oplus \mathbb{1}[\mathcal{Y}]$, where $\mathbb{1}[\mathcal{Y}]$ is a mask of all 1's for each frame indexed in $\mathcal{Y}$. This masks only the missing pixels in frames $\mathcal{X}$ while treating all pixels in frames $\mathcal{Y}$ as observed (visualized in Fig. 4). We then extract our training targets from the video as $\mathbf{x} := \mathbf{V}[\mathcal{X} \oplus \mathcal{Y}] \odot (1 - \mathbf{M}_{\mathcal{X},\mathcal{Y}})$, and our observations as $\mathbf{y} := \mathbf{V}[\mathcal{X} \oplus \mathcal{Y}] \odot \mathbf{M}_{\mathcal{X},\mathcal{Y}}$.

**Training Objective.** Sampling $\mathbf{V}$, $\mathbf{M}$, $\mathcal{X}$, and $\mathcal{Y}$ for every training example therefore defines a distribution over $\mathbf{x}$ and $\mathbf{y}$, which we use when estimating the expectation over them in Eq. (2). Combining this method of sampling $\mathbf{x}$ and $\mathbf{y}$ with our pixel-wise mask, we write the loss as

$$\mathcal{L}(\theta) := \mathbb{E}_{t,\mathbf{x},\mathbf{y},\mathcal{X},\mathcal{Y},\epsilon} \left[ (1 - \mathbf{M}_{\mathcal{X},\mathcal{Y}}) \odot \left\| \epsilon - \epsilon_\theta \left( \sqrt{\bar{\alpha}_t} \mathbf{x} + \sqrt{1 - \bar{\alpha}_t} \epsilon, \mathbf{y}, \mathbf{M}_{\mathcal{X},\mathcal{Y}}, \mathcal{X}, \mathcal{Y}, t \right) \right\|^2 \right], \quad (3)$$

where $\mathcal{X}, \mathcal{Y}$ provide information about each frame's index within $\mathbf{V}$, and $\mathbf{M}_{\mathcal{X},\mathcal{Y}}$ is the mask. Note that the loss is only computed for missing pixels, as indicated by the elementwise multiplication of the loss by $(1 - \mathbf{M}_{\mathcal{X},\mathcal{Y}})$.

---

[1]Our frame index distribution is a mixture distribution between the one used by FDM and one which always samples $\mathcal{X}$ and $\mathcal{Y}$ so that they represent sequences of consecutive frames. We found that including these sequences of consecutive frames improved temporal coherence.

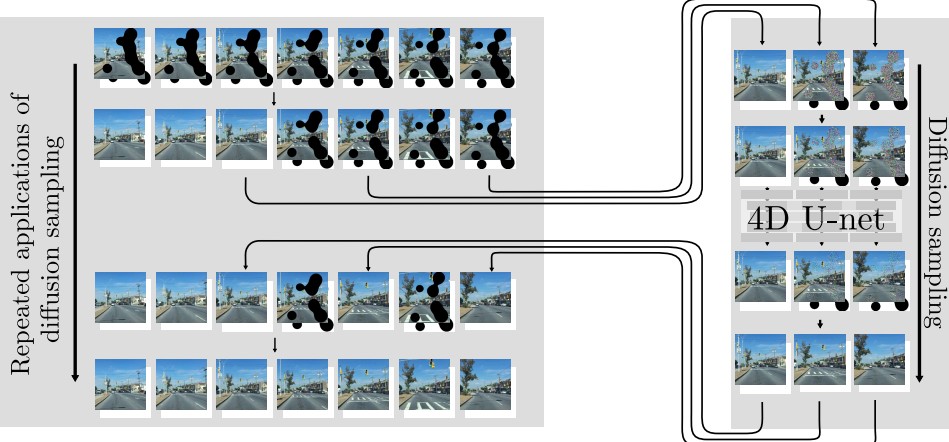

Figure 5: A visualization of our inpainting procedure using an arbitrary sampling scheme. At each *sampling stage*, a subsequence of frames and their corresponding masks are selected to be generated and conditioned on. At each step in the inner denoising process, only the pixels corresponding to masked regions are updated, leaving ground truth and previously inpainted pixels unaltered.

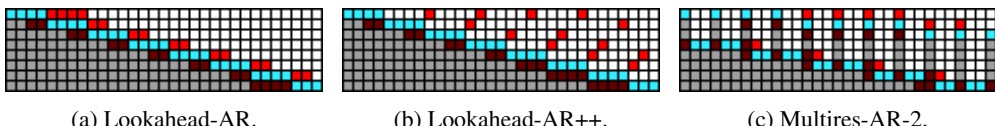

(a) Lookahead-AR.          (b) Lookahead-AR++.          (c) Multires-AR-2.

Figure 6: Sampling schemes visualizations similar to Fig. 3. In addition we now also condition on observed pixel values in frames that can also contain unknown pixel values. Frames where we do so are shown in bright red and the color scheme is otherwise the same as in Fig. 3.

**Sampling.** Sampling from a model trained using the above objective can be done with any diffusion model sampler. The only difference is that, at each step in the reverse process, only the values of the unknown pixels are updated, such that the known pixels retain their ground-truth values at each step. We use the Heun sampler proposed by Karras et al. (2022), and leave further details to the appendix.

### 4.3 CONDITIONING ON INCOMPLETE FRAMES

Given our proposed architecture and training procedure, we can sample from the resulting models using the FDM sampling schemes shown in Fig. 3 without further complication. A downside of these sampling schemes, however, is that they do not enable us to condition on frames with unknown pixel values. That is, we are unable to condition on the known pixels in a frame unless we either (a) have previously inpainted it and know all of its pixel values already, or (b) are inpainting it as we condition on it. We show in our experiments that this often leaves us unable to account for important dependencies in the context.

We therefore propose a method for conditioning on *incomplete* frames. This enables the sampling schemes shown in Fig. 6, where we condition on the known pixels in incomplete frames, marked in red. Recall that $\mathbf{x}$ denotes "unknown pixels in frames indexed by $\mathcal{X}$" and $\mathbf{y}$ denotes "known pixels in frames indexed by $\mathcal{X}$ or $\mathcal{Y}$". If any frames indexed by $\mathcal{Y}$ are incomplete then we have a third category, which we'll call $\mathbf{z}$: "unknown pixels in frames indexed by $\mathcal{Y}$".

We then wish to approximately sample $\mathbf{x} \sim p_{\text{data}}(\cdot|\mathbf{y})$ without requiring values of $\mathbf{z}$. We do not have a way to sample directly from an approximation of this distribution, as the diffusion model is not trained to condition on "incomplete" frames. We note, however, that this desired distribution is the marginal of a distribution that our diffusion model *can* approximate:

$$p_{\text{data}}(\mathbf{x}|\mathbf{y}) = \int p_{\text{data}}(\mathbf{x}, \mathbf{z}|\mathbf{y})\mathrm{d}\mathbf{z} \approx \int p_{\theta}(\mathbf{x}, \mathbf{z}|\mathbf{y})\mathrm{d}\mathbf{z}. \tag{4}$$

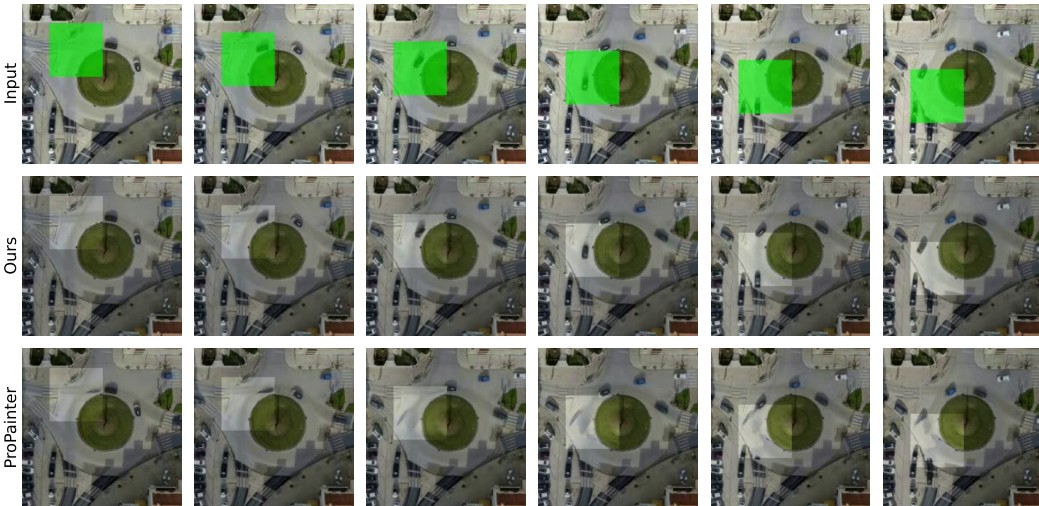

Figure 7: Inpaintings from our model and ProPainter on the task introduced in Fig. 2 from our Traffic-Scenes dataset. Our model can inpaint a realistic trajectory for the occluded vehicle. Competing flow-based approaches correctly inpaint the background but are unable to account for the vehicle.

This means that we can sample from the required approximation of $p_{\text{data}}(\mathbf{x}|\mathbf{y})$ by sampling from $p_\theta(\mathbf{x}, \mathbf{z}|\mathbf{y})$ and then simply discarding $\mathbf{z}$.

### 4.4 SAMPLING SCHEMES FOR VIDEO INPAINTING

The ability to condition on incomplete frames enables us to design new sampling schemes that better capture dependencies that are necessary for high-quality video inpainting. **Lookahead-AR** is a variant of AR that takes into account information from future frames by conditioning on the observed parts of the frames immediately after the sequence of frames being generated, as well as on the frames before; see Fig. 6a. **Lookahead-AR++** builds on "Lookahead-AR" by conditioning on the observed parts of frames far in the future instead of of frames immediately after those being sampled; see Fig. 6b. **Multires-AR-3** builds on "Lookahead-AR" by first inpainting every fifteenth frame using Lookahead-AR, then inpainting every fifth frame while conditioning on nearby inpainted frames, and then inpainting all other frames. We visualize a "Multires-AR-2" version (inpainting every third frame and then every frame) in Fig. 6c. Visualizations of all sampling schemes considered in this work can be found in Section D.

## 5 EXPERIMENTS

### 5.1 DATASETS

To highlight the unique capabilities our generative approach offers, we wish to target video inpainting tasks in which visual information in nearby frames cannot be easily exploited to achieve a convincing result. The YouTube-VOS (Xu et al., 2018) (training and test) and DAVIS (Perazzi et al., 2016) (test) video object segmentation datasets have become the *de facto* standard benchmark datasets for video inpainting in recent years, and the foreground object masks included in these datasets have led to a heavy focus on object removal tasks in qualitative evaluations of video inpainting methods. Object removal in these datasets is a task to which content propagation methods are specifically well-suited, as the backgrounds are often near-stationary and so information in neighboring frames can be used to great effect. In contrast, we wish to focus on tasks where inpainting requires the visual appearances of objects to be hallucinated in whole or in part and realistically propagated through time, or where the behavior of occluded objects must be inferred. We propose three new large-scale video inpainting datasets targeting such tasks. All datasets are $256 \times 256$ res-

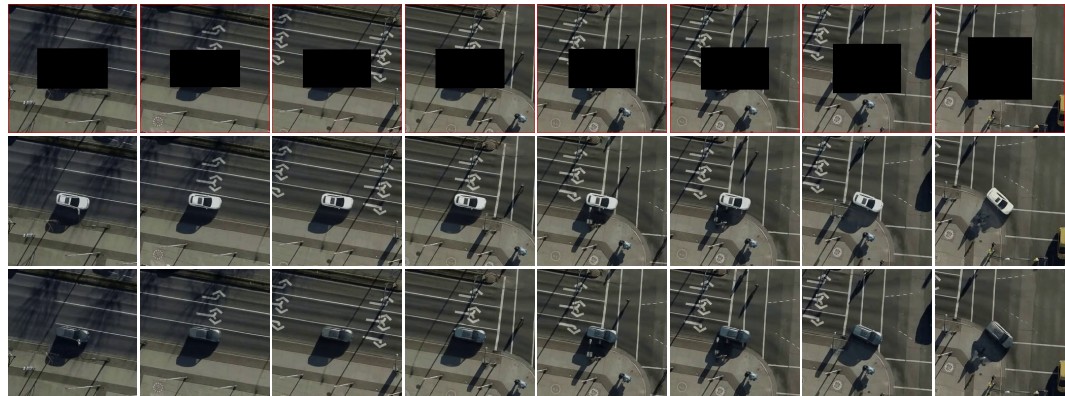

Figure 8: Inpaintings from our model on the Inpainting-Cars dataset. The first row is the masked input to the model, the lower two rows are two separate inpaintings sampled from our model.

olution and 10 fps. Representative examples of each dataset, along with details of how each dataset was constructed, are included in Section F.

**BDD-Inpainting.** We adapt the BDD100K (Yu et al., 2020) dataset for video inpainting. This dataset contains 100,000 high-quality first-person driving videos and includes a diversity of geographic locations and weather conditions. We select a random subset of approximately 50,000 of these, and generate a set of both moving and stationary masks of four types: grids, horizontal or vertical lines, boxes (Fig. 7), and blobs (Fig. 1). During training both videos and masks are sampled uniformly, giving a distribution over an extremely varied set of inpainting tasks. We include two test sets of 100 video-mask pairs each, the first (BDD-Inpainting) containing only grid, line, and box masks, and the second (BDD-Inpainting-Blobs) containing only blob masks. The second is a substantially more challenging test set, as the masks are often large and occlude objects in the scene for a significant period of time. See the appendix for representative examples; mask types, shown in green, are indicated for each. The test sets we use, along with code for processing the original BDD100K dataset and generating masks, will be released upon publication.

**Inpainting-Cars.** We use an in-house dataset of overhead drone footage of vehicle traffic to create a dataset targeting the *addition* of cars to videos. Crops are taken such that each video centers a single vehicle, and tracker-generated bounding boxes are used as masks. Note that the masked-out vehicle is not visible at any time in the context, and so, given an input, the model must generate a vehicle and propagate it through time, using only the size and movement of the mask as clues to plausible vehicle appearance and motion.

**Traffic Scenes.** Using the same in-house dataset as referenced above, we create another dataset where large sections of the roadway are occluded, requiring the model to infer vehicle behavior in the masked-out region. Crops are centered over road features like intersections, roundabouts, highway on-ramps, *etc.* where complex vehicle behavior and interactions take place. This dataset contains exceptionally challenging video inpainting tasks as vehicles are often occluded for long durations of time, requiring the model to generate plausible trajectories that are semantically consistent with both the observations (e.g. the entry and exit point of a vehicle from the masked region) and the roadway. This dataset contains approximately 13,000 videos in the training set and 100 video-mask pairs in the test set.

## 5.2 BASELINES AND METRICS

We compare our method with four recently proposed video inpainting methods that achieve state-of-the-art performance on standard benchmarks: ProPainter (Zhou et al., 2023), E$^2$FGVI (Li et al., 2022), FGT (Zhang et al., 2022), and FGVC (Gao et al., 2020). For each model, we use pre-trained checkpoints made available by the respective authors. We adopt the evaluation suite of DEVIL (Szeto & Corso, 2021), which includes a comprehensive selection of commonly used metrics targeting three different aspects of inpainting quality:

Table 1: Quantitative comparison with state-of-the-art video inpainting methods across each of our four datasets. For each dataset/metric, the best performing model is indicated with bold font and the second best performing model is underlined. Lookahead-AR++ is used for all datasets.

| Method | PSNR▲ | SSIM▲ | LPIPS▼ | PVCS▼ | FID▼ | VFID▼ | $E_{\text{warp}}$▼ |
|---|---|---|---|---|---|---|---|
| | | | BDD-Inpainting | | | | |
| ProPainter | 32.65 | 0.968 | 0.0355 | 0.2806 | 2.51 | 0.1599 | **1.5·10⁻³** |
| FGT | 28.50 | 0.928 | 0.0843 | 0.6430 | 9.94 | 0.7863 | 4.0·10⁻³ |
| E²FGVI | 30.16 | 0.946 | 0.0640 | 0.4765 | 6.58 | 0.3665 | 2.3·10⁻³ |
| FGVC | 25.84 | 0.884 | 0.1498 | 1.0210 | 32.08 | 1.7995 | 7.5·10⁻³ |
| Ours | **33.68** | **0.972** | **0.0261** | **0.2037** | **1.71** | **0.0748** | 1.8·10⁻³ |
| | | | BDD-Inpainting-Blobs | | | | |
| ProPainter | 30.54 | 0.960 | 0.0467 | 0.3120 | 2.56 | 0.1499 | **1.2·10⁻³** |
| FGT | 27.33 | 0.938 | 0.0737 | 0.5130 | 9.41 | 0.3594 | 2.5·10⁻³ |
| E²FGVI | 29.04 | 0.950 | 0.0667 | 0.4414 | 4.23 | 0.2339 | 1.7·10⁻³ |
| FGVC | 25.10 | 0.913 | 0.0980 | 0.6957 | 16.64 | 0.6184 | 3.8·10⁻³ |
| Ours | **30.67** | **0.961** | **0.0442** | **0.2857** | **1.69** | **0.1083** | 1.5·10⁻³ |
| | | | Traffic-Scenes | | | | |
| ProPainter | 31.98 | 0.967 | 0.0326 | 0.2909 | 8.51 | 0.3482 | 2.5·10⁻⁴ |
| FGT | 31.97 | 0.963 | 0.0397 | 0.3528 | 9.92 | 0.5392 | 3.6·10⁻⁴ |
| E²FGVI | 31.40 | 0.957 | 0.0440 | 0.3809 | 13.13 | 0.6113 | 4.1·10⁻⁴ |
| FGVC | 29.11 | 0.926 | 0.0794 | 0.5914 | 25.93 | 1.2358 | 9.9·10⁻⁴ |
| Ours | **35.29** | **0.978** | **0.0202** | **0.1725** | **4.87** | **0.1637** | **2.3·10⁻⁴** |

- Reconstruction, or how well the method's output matches the ground truth: PSNR, SSIM (Wang et al., 2004), LPIPS (Zhang et al., 2018a), PVCS (Szeto & Corso, 2021).

- Perceptual realism, or how well the appearance and/or motion resembles a reference set of ground truth videos: FID (Heusel et al., 2017), VFID (Wang et al., 2018).

- Temporal consistency: Flow warping error ($E_{\text{warp}}$) (Lai et al., 2018), which measures how well an inpainting follows the optical flow as calculated on the ground truth.

For our method, we train one model on each dataset with $K = 16$. Models are trained on $4\times$ NVIDIA A100 GPUs for 1-4 weeks. A detailed accounting of each model's hyperparameters and training procedure can be found in the appendix.

## 5.3 QUANTITATIVE EVALUATION

We report quantitative results for three of our datasets in Table 1. For each dataset, for our method we report metrics for the best-performing model and sampling scheme we found for that dataset. Our method outperforms the baselines on all metrics except $E_{\text{warp}}$ for all datasets, often by a significant margin. We suspect the discrepancy between $E_{\text{warp}}$ and the other metrics arises because each of the competing methods predicts a completion of the optical flow field and utilizes this during the inpainting process, in a sense explicitly targeting $E_{\text{warp}}$. Inpainting-Cars is omitted from this section, as we are not aware of an existing method suitable for this task.

## 5.4 QUALITATIVE EVALUATION

**BDD-Inpainting.** On this dataset, qualitative differences between our method and competing approaches are most evident in the presence of large masks that partially or fully occlude objects for a long period of time. In such cases our method can retain or complete the visual appearance of occluded objects and propagate them through time in a realistic way, while such objects tend to disappear gradually with flow-based approaches (as in Fig. 1). Qualitative results on this dataset are heavily influenced by the sampling scheme used. Our "Lookahead-AR++" sampling scheme tends to perform best, as it allows information from both past and future frames to be conditioned on in the inpainting process. See Section H.2 for a qualitative demonstration of the effects of different sampling schemes. Additional results are provided in Section J.1.

**Traffic-Scenes.** On this dataset, all competing methods tend to inpaint only the road surface; when vehicles enter the occluded region they disappear almost instantaneously. Our method shows a surprising ability to inpaint long, realistic trajectories that are consistent with the entry/exit points of

Table 2: Effect of sampling schemes, measured on the Traffic-Scenes test set.

| Sampling Scheme | PSNR▲ | SSIM▲ | LPIPS▼ | PVCS▼ | FID▼ | VFID▼ | $E_{\text{warp}}$▼ |
|---|---|---|---|---|---|---|---|
| AR (Harvey et al., 2022) | 31.72 | 0.9595 | 0.0392 | 0.2989 | 8.85 | 0.3016 | $3.01 \cdot 10^{-4}$ |
| Hierarchy-2 (Harvey et al., 2022) | **35.53** | **0.9794** | **0.0196** | 0.1732 | **4.55** | 0.1714 | $2.48 \cdot 10^{-4}$ |
| Lookahead-AR++ (ours) | 35.29 | 0.9783 | 0.0202 | **0.1725** | 4.87 | **0.1637** | **$2.26 \cdot 10^{-4}$** |
| Multires-AR-3 (ours) | 35.32 | 0.9785 | 0.0210 | 0.1815 | 5.07 | 0.1821 | $2.43 \cdot 10^{-4}$ |

Table 3: Effect of diffusion samplers, using AR sampling on BDD-Inpainting.

| Sampler | PSNR▲ | SSIM▲ | LPIPS▼ | PVCS▼ | FID▼ | VFID▼ | $E_{\text{warp}}$▼ |
|---|---|---|---|---|---|---|---|
| Heun (10 steps) | 33.00 | 0.9687 | 0.0339 | 0.2437 | 2.39 | 0.1155 | **$1.66 \cdot 10^{-3}$** |
| Heun (25 steps) | **33.05** | **0.9703** | 0.0290 | 0.2202 | 1.84 | 0.0815 | $1.78 \cdot 10^{-3}$ |
| Heun (50 steps) | 33.00 | 0.9699 | 0.0282 | 0.2177 | 1.81 | **0.0766** | $1.82 \cdot 10^{-3}$ |
| Heun (100 steps) | 32.98 | 0.9700 | 0.0280 | **0.2171** | 1.76 | 0.0788 | $1.85 \cdot 10^{-3}$ |
| DDPM (1000 steps) | 32.41 | 0.9665 | **0.0272** | 0.2244 | **1.68** | 0.0779 | $2.39 \cdot 10^{-3}$ |

vehicles into the occluded region, as well as correctly following roadways as in Fig. 7. Additional results are provided in Section J.2.

**Inpainting-Cars.** Sampled inpaintings from our model on an example from the Inpainting-Cars dataset are shown in Fig. 8. Our model is capable of generating diverse, visually realistic inpaintings for a given input. We note the model's ability to use *semantic* cues from the context to generate plausible visual features, such as realistic behavior of shadows and plausible turning trajectories based only on the movement of the mask. Additional results are provided in Section J.3.

### 5.5 ABLATIONS

**Effect of Sampling Schemes.** We compare selected sampling schemes on the Traffic-Scenes dataset in Table 2, before providing more thorough qualitative and quantitative comparisons in the appendix. We find Lookahead-AR++ performs best on all datasets in terms of video realism (VFID) and temporal consistency (warp error), demonstrating the importance of our method's ability to condition on incomplete frames. AR tends to do poorly on all metrics on both the Traffic-Scenes and BDD-Inpainting datasets; this is likely due to its inability to account for observed pixel values in frames coming after the ones being generated at each stage, causing divergence from the ground-truth and then artifacts when producing later frames conditioned on incompatible values. Hierarchy-2 does well on some metrics of reconstruction and frame-wise perceptual quality. It does less well in terms of video realism, as the inpainting of the "keyframes" in the first stage is done without conditioning on the known pixels of frames in their immediate vicinity, potentially leading to incompatibility with the local context.

**Samplers and Number of Sampling Steps.** The use of diffusion models allows for a trade-off between computation time and inpainting quality by varying the sampler and number of steps used in the generative process. In Table 3 we report the results of our model on the BDD-Inpainting test set with the AR sampling scheme using different samplers and numbers of sampling steps. Aligning with expectations and qualitative observations, performance on metrics degrades as the number of sampling steps decreases for LPIPS, PVCS, FID and VFID. We note, however, that performance on $E_{\text{warp}}$, PSNR, and SSIM improves, suggesting that these metrics may not correlate well with perceived quality, as has been previously suggested for the latter two metrics in Zhang et al. (2018b).

## 6 CONCLUSION

In this work, we have presented a framework for video inpainting by conditioning video diffusion models. This framework allows for flexible conditioning on the context, enabling post-hoc experimentation with sampling schemes which can greatly improve results. We introduce four challenging tasks for video inpainting methods and demonstrate our model's ability to use semantic information in the context to solve these tasks effectively. Our experiments demonstrate a clear improvement on quantitative metrics and, contrary to existing methods, our approach can generate semantically meaningful completions based on minimal information in the context.

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

# A LIMITATIONS

The most notable limitation of our method is its computational cost relative to competing methods. The wall-clock time for our method is highly dependent on a number of factors, such as the number of sampler steps used and the number of frames generated in each stage of the sampling scheme used. See Section C for further details. Improvements in few-step generation for diffusion models and increasing the size of our model such that more frames can be processed at a time would both help to mitigate these concerns. Additionally, our method requires that a model be trained on a specific dataset which is reasonably close to the data distribution to be inpainted at test time, requiring large datasets. The development of large scale, general purpose video generative models would likely help to make generative approaches such as ours more robust to out-of-distribution tasks.

# B POTENTIAL NEGATIVE IMPACTS

The development of generative models for the generation of photo-realistic video opens the door to malicious or otherwise unethical uses which could have negative impacts on society. While our method is currently limited to operating on data which is sufficiently similar to the datasets it was trained on, and those datasets have limited potential for malicious use, future work which takes a generative approach to video inpainting could be used for the purposes of misinformation, harassment or deception.

# C COMPUTATIONAL COSTS

Below we include an accounting of the computational cost of our method relative to the baselines we considered for inpainting two examples. We decided to include two examples as the inference costs for the methods that we compare against depend not only on the number of frames but also on the mask and content of the videos being inpainted, and can vary significantly between videos. For FGVC we include results both with and without the "seamless" flag, which the authors claim produces superior results at the cost of increased processing time. Note that the results for FGVC reported in the paper are computed using the "seamless" option. Additionally, we include inference costs for our method using both 100 and 10 Heun sampler steps, recalling that our method allows for a tradeoff between computational cost and inpainting quality. While our method has the highest memory requirement out of those we compare to, it remains competitive in terms of runtime, particularly using fewer sampling steps.

Table 4: Runtime and peak GPU memory usage for all methods on the example shown in Fig. 1.

| Method | Runtime | Peak GPU Memory (GB) |
|---|---|---|
| ProPainter | 1m7s | 2.68 |
| E2FGVI | 0m54s | 3.47 |
| FGVC (seamless) | 228m43s | 0.13 |
| FGVC | 18m48s | 0.13 |
| FGT | 6m27s | 4.23 |
| Ours (100 steps) | 9m22s | 7.29 |
| Ours (10 steps) | 1m6s | 7.29 |

Table 5: Runtime and peak GPU memory usage for all methods on the example shown in Fig. 15.

| Method | Runtime | Peak GPU Memory (GB) |
|---|---|---|
| ProPainter | 0m32s | 2.84 |
| E2FGVI | 0m19s | 3.47 |
| FGVC (seamless) | 15m27s | 0.13 |
| FGVC | 4m36s | 0.13 |
| FGT | 11m19s | 4.23 |
| Ours (100 steps) | 9m58s | 7.29 |
| Ours (10 steps) | 1m6s | 7.29 |

## D  SAMPLING SCHEME DETAILS

Figure 9 depicts the sampling schemes used in all experiments for a video length of 200 to supplement the descriptions given in the main text.

## E  ALGORITHMS

See Algorithm 1 for our training procedure and Algorithm 2 for our sampling procedure.

---

**Algorithm 1** Training Loop

---

1: **repeat**
2:     $(\mathbf{V}, \mathbf{M}) \sim q(\mathbf{V}, \mathbf{M})$         ▷ Sample video and mask
3:     $(\mathcal{X}, \mathcal{Y}) \sim u(\mathcal{X}, \mathcal{Y})$         ▷ Sample training task
4:     $t \sim \mathcal{U}(\{1 \ldots T\})$
5:     $\boldsymbol{\epsilon} \sim \mathcal{N}(\mathbf{0}, \mathbf{I})$
6:     $\mathbf{M}_{\mathcal{X},\mathcal{Y}} \leftarrow \mathbf{M}[\mathcal{X}] \oplus \mathbb{1}[\mathcal{Y}]$         ▷ Construct input mask
7:     $(\mathbf{x}, \mathbf{y}) \leftarrow (\mathbf{V}[\mathcal{X} \oplus \mathcal{Y}] \odot (1 - \mathbf{M}_{\mathcal{X},\mathcal{Y}}), \mathbf{V}[\mathcal{X} \oplus \mathcal{Y}] \odot \mathbf{M}_{\mathcal{X},\mathcal{Y}})$    ▷ Extract $\mathbf{x}$ and $\mathbf{y}$
8:     Take gradient descent step on objective defined in Eq. (3),
$$\nabla_\theta \left\| \boldsymbol{\epsilon} - \epsilon_\theta \left(\sqrt{\bar{\alpha}_t}\mathbf{x} + \sqrt{1 - \bar{\alpha}_t}\boldsymbol{\epsilon}, \mathbf{y}, \mathbf{M}_{\mathcal{X},\mathcal{Y}}, \mathcal{X}, \mathcal{Y}, t\right) \right\|^2$$    ▷ Masked loss
9: **until** converged

---

---

**Algorithm 2** Inpaint video $\mathbf{V}$ given mask $\mathbf{M}$ and sampling scheme $[(\mathcal{X}_s, \mathcal{Y}_s)]_{s=1}^{S}$. Note here that $\mathcal{Y}_s$ may contain incomplete frames, as discussed in Section 4.3. maskedDDPM$(\cdot; \ldots)$ denotes a generative diffusion process where only the masked pixels are updated at each denoising step, as illustrated in Fig. 5.

---

1: **for** $s \leftarrow 1, \ldots, S$ **do**
2:     $\mathbf{V}_s \leftarrow \mathbf{V}[\mathcal{X}_s \oplus \mathcal{Y}_s]$         ▷ Extract selected video frames
3:     $\mathbf{M}_s \leftarrow \mathbf{M}[\mathcal{X}_s \oplus \mathcal{Y}_s]$         ▷ Extract selected mask frames
4:     $\hat{\mathbf{V}}_s \sim \texttt{maskedDDPM}(\cdot; \mathbf{V}_s, \mathbf{M}_s, \mathcal{X}, \mathcal{Y}, \theta)$         ▷ Sample inpainting for subvideo
5:     $\mathbf{V}[\mathcal{X}_s] \leftarrow \hat{\mathbf{V}}_s$         ▷ Insert completed frames at indices $\mathcal{X}_s$
6:     $\mathbf{M}[\mathcal{X}_s] \leftarrow \mathbb{1}$         ▷ Update masks for inpainted frames
7: **end for**
8: **return** $\mathbf{V}$

---

## F  DATASET DETAILS

In Section F.1 we detail how our datasets were generated. Section F.2 shows representative examples from each of our datasets.

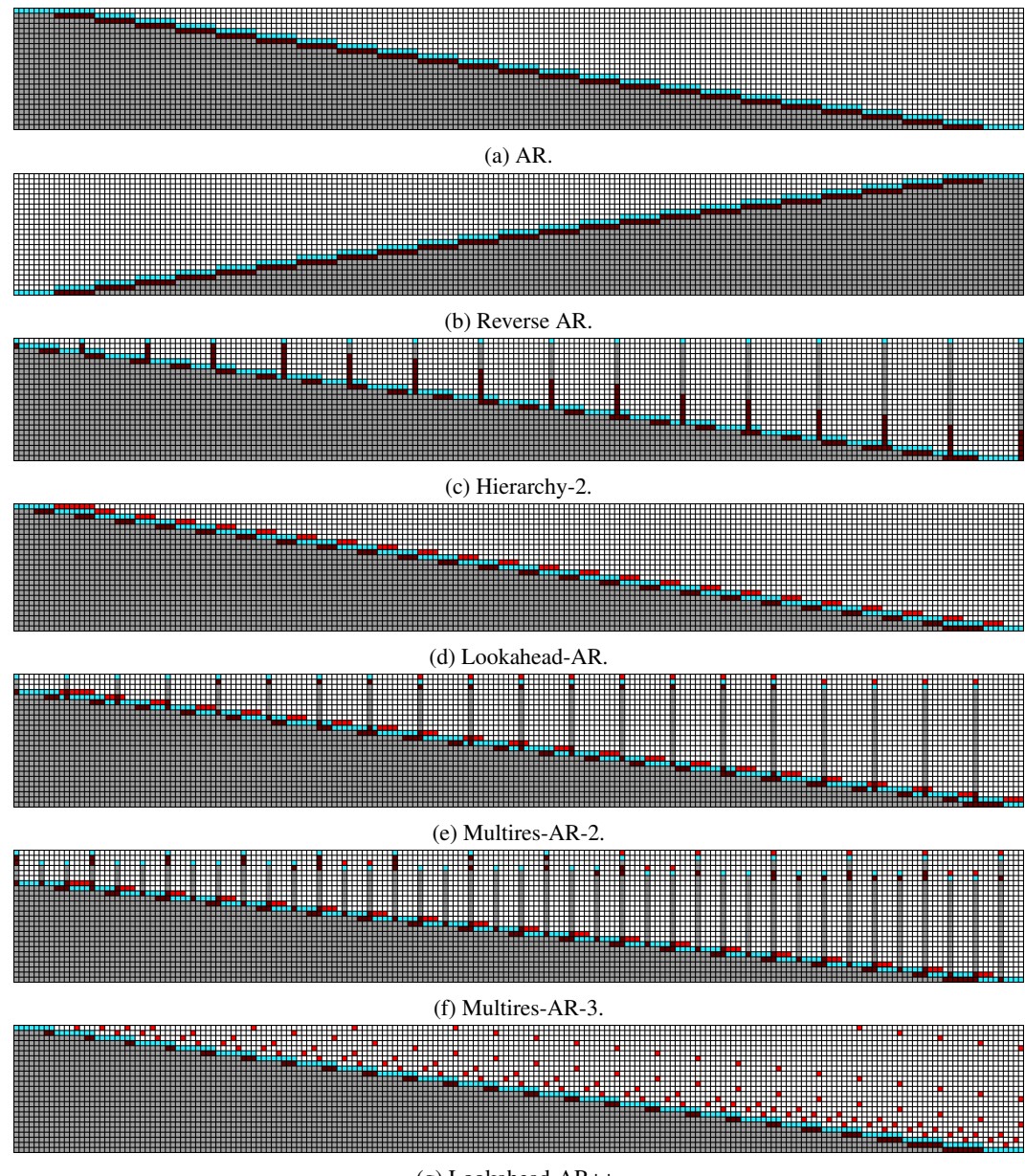

(a) AR.

(b) Reverse AR.

(c) Hierarchy-2.

(d) Lookahead-AR.

(e) Multires-AR-2.

(f) Multires-AR-3.

(g) Lookahead-AR++.

Figure 9: Sampling schemes visualizations similar to Fig. 3 but with a model capable of attending to 16 frames at a time, as in our experiments, and a more typical video length of 200.

### F.1 DATASET CREATION

#### F.1.1 BDD-INPAINTING

From the original BDD100K dataset we take a random subset of videos, using 48,886 for a training set and 100 for each held-out test set. Videos are downsampled spatially by a factor of 2.5, center-cropped to $256 \times 256$, and the frame rate is reduced by a factor of three to 10 fps. We truncate videos to 400 frames, corresponding to a 40 second length for each. We randomly generate a set of 49,970 masks of four types: grids, horizontal or vertical lines, boxes, and blobs. Our mask generation procedure first picks a mask type, and then randomly selects mask parameters such as size and direction of motion (which includes stationary masks). Generated masks contain 400 frames. During training both videos and masks are sampled uniformly and independently, giving a distribution over

nearly 2.5 billion video-mask pairs. Our test sets, as well as code for regenerating our training set, will be made public upon publication.

### F.1.2 INPAINTING-CARS

We use an in-house dataset of overhead drone footage of vehicle traffic, for which we have tracker-generated bounding boxes for each vehicle. The videos are spatially downsampled by a factor of two from their original 4k resolution, and $256 \times 256$ sub-videos centred on vehicles are extracted using the vehicle-specific tracks. The length of these videos is variable as it depends on the amount of time a given vehicle was visible in the source video, but typically ranges from 10 to 60 seconds at 10 fps. Bounding boxes are dilated by a factor of two to ensure the entirety of the vehicle is contained within them, and these dilated bounding boxes are then used as masks. This dataset contains 2973 training examples and 100 held-out test examples.

### F.1.3 TRAFFIC SCENES

This dataset is created using the same in-house dataset as was used for Inpainting-Cars. The 4k, 10 fps source videos are spatially downsampled by a factor of 7.25, truncated to 200 frames and cropped to $256 \times 256$, with the crops centred over road features like intersections, roundabouts, highway on-ramps, *etc*. We generate masks using the same mask generation procedure as in BDD-Inpainting.

### F.2 DATASET EXAMPLES

### F.2.1 BDD-INPAINTING

See Fig. 10 for examples from the BDD-Inpainting dataset. Mask types, shown in green, are indicated for each example.

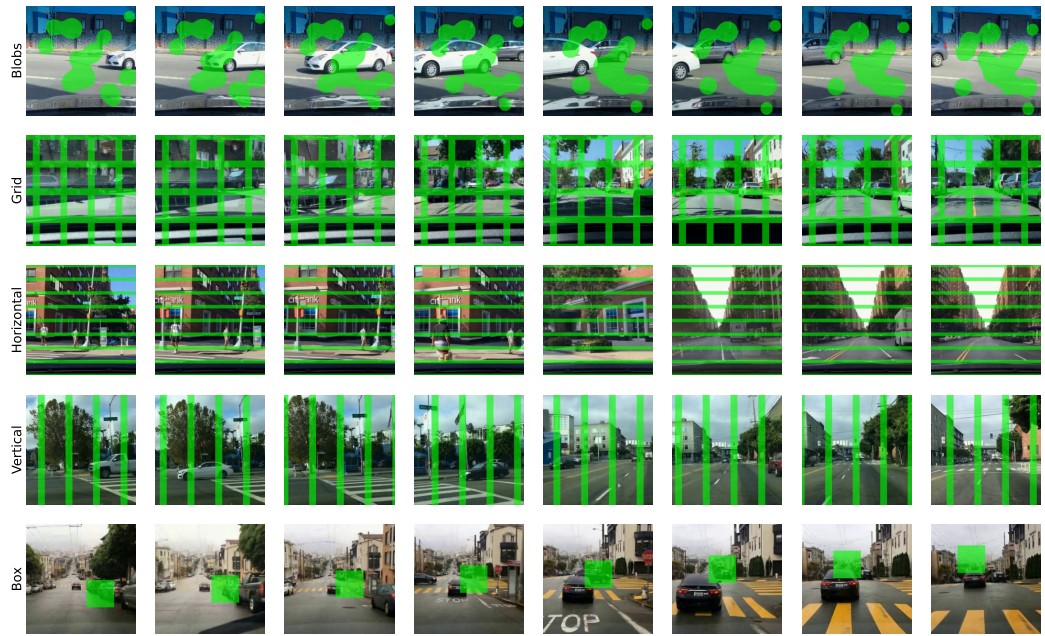

Figure 10: Representative examples from the BDD-Inpainting dataset.

### F.2.2 INPAINTING-CARS

See Fig. 11 for examples from the Inpainting-Cars dataset. Masks are shown in black.

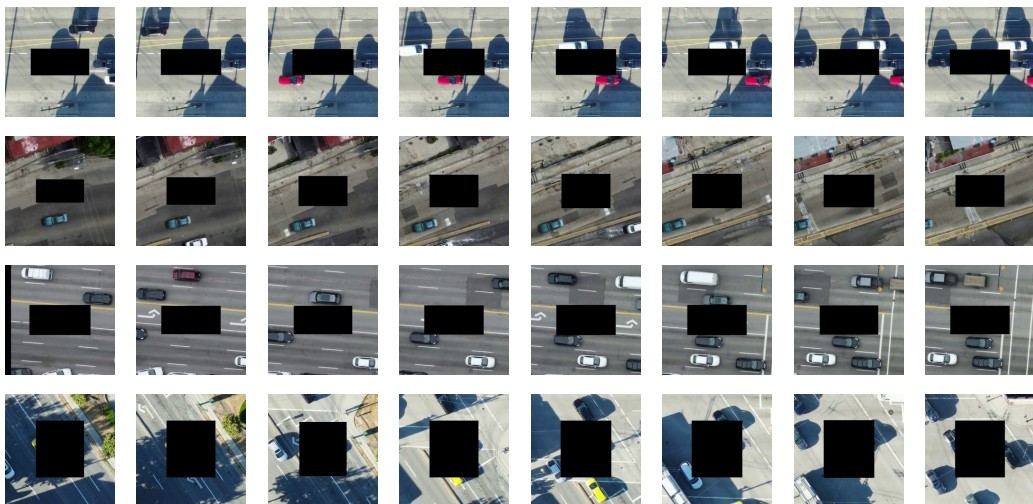

Figure 11: Representative examples from the Inpainting-Cars dataset.

### F.2.3 TRAFFIC-SCENES

See Fig. 12 for examples from the BDD-Inpainting dataset. Mask types, shown in green, are indicated for each example.

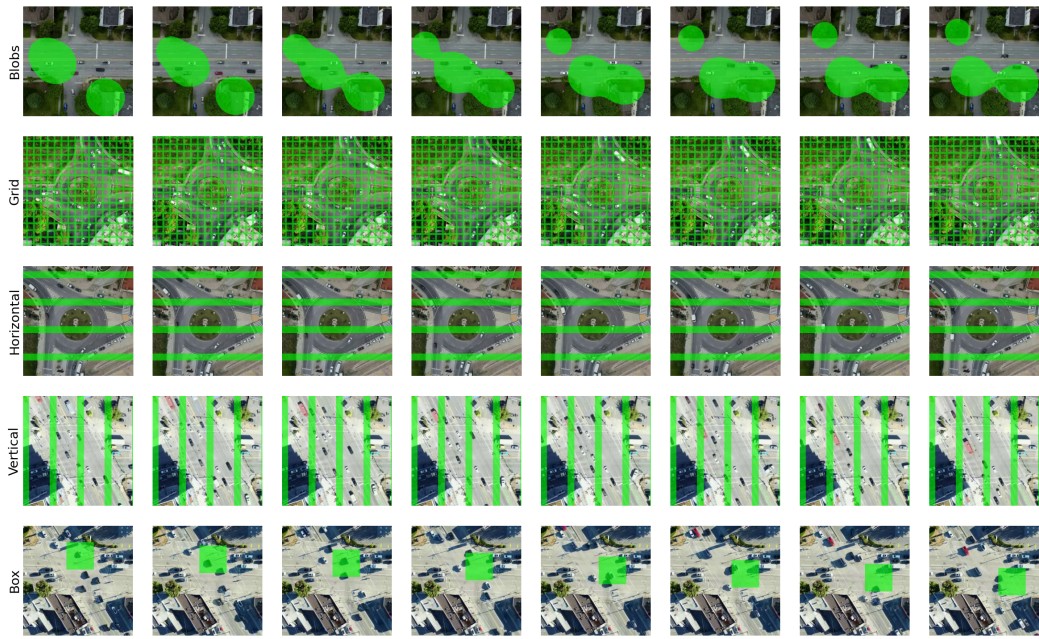

Figure 12: Representative examples from the Traffic-Scenes dataset.

## G TRAINING DETAILS

All models are trained with $4 \times$ NVIDIA A100 GPUS with a batch size of one, corresponding to an effective batch size of 4 as gradients are aggregated across GPUs. All models were trained to condition on or generate 16 frames at a time, use an EMA rate of 0.999, and use the AdamW

Loshchilov & Hutter (2019) optimizer. We use the noise schedules defined in Chen Chen (2023): `cosine` for Inpainting-Cars, and `sigmoid` for BDD-Inpainting and Traffic-Scenes. All models use temporal attention at the spatial resolutions (32, 16, 8) in the U-Net, except for the Inpainting-Cars model where temporal attention only occurs at resolutions (16, 8). Otherwise, the default hyperparameters from our training code are used for all models. This code, and the BDD-Inpainting checkpoint, will be released upon publication. The number of training iterations for each model is listed below:

| Dataset | Iterations (millions) |
|---|---|
| BDD-Inpainting | 2.5 |
| Inpainting-Cars | 1.1 |
| Traffic-Scenes | 2.4 |

# H    ADDITIONAL ABLATIONS

## H.1    ADDITIONAL SAMPLING SCHEME ABLATIONS

Table 6 and Table 7 show the effect of using different sampling schemes on the BDD-Inpainting and BDD-Inpainting-Blobs test sets, respectively. The sampling schemes tested are depicted in Fig. 3 and Fig. 6. For each metric, the best performing sampling scheme is indicated in bold font. All measurements were taken using the Heun sampler Karras et al. (2022) with 100 sampling steps. The Lookahead-AR++ performs best on both test sets across almost all metrics. Sampling scheme ablations are omitted for Inpainting-Cars as we found an autoregressive scheme gave significantly better qualitative results than the other schemes.

Table 6: Effect of sampling schemes measured on the BDD-Inpainting test set.

| Sampling Scheme | PSNR▲ | SSIM▲ | LPIPS▼ | PVCS▼ | FID▼ | VFID▼ | $E_{\text{warp}}$▼ |
|---|---|---|---|---|---|---|---|
| Multires-AR-3 | 32.81 | 0.9678 | 0.0289 | 0.2302 | 1.75 | 0.0884 | $2.19 \cdot 10^{-3}$ |
| Lookahead-AR++ | 33.68 | **0.9717** | **0.0261** | **0.2037** | **1.71** | **0.0748** | $\mathbf{1.79 \cdot 10^{-3}}$ |
| AR | 32.97 | 0.9699 | 0.0278 | 0.2166 | 1.78 | 0.0778 | $1.85 \cdot 10^{-3}$ |
| Hierarchy-2 | 32.96 | 0.9690 | 0.0284 | 0.2232 | 1.74 | 0.0839 | $1.97 \cdot 10^{-3}$ |
| Multires-AR-2 | 33.18 | 0.9692 | 0.0278 | 0.2201 | 1.72 | 0.0815 | $2.03 \cdot 10^{-3}$ |
| Reverse AR | **33.31** | 0.9702 | 0.0273 | 0.2132 | 1.76 | 0.0785 | $\mathbf{1.79 \cdot 10^{-3}}$ |

Table 7: Effect of sampling schemes measured on the BDD-Inpainting-Blobs test set.

| Sampling Scheme | PSNR▲ | SSIM▲ | LPIPS▼ | PVCS▼ | FID▼ | VFID▼ | $E_{\text{warp}}$▼ |
|---|---|---|---|---|---|---|---|
| Multires-AR-3 | 29.89 | 0.9561 | 0.0475 | 0.3142 | 1.63 | 0.1188 | $2.00 \cdot 10^{-3}$ |
| Lookahead-AR++ | **30.67** | **0.9608** | **0.0442** | **0.2857** | 1.69 | **0.1083** | $1.53 \cdot 10^{-3}$ |
| AR | 29.45 | 0.9547 | 0.0512 | 0.3319 | 2.07 | 0.1328 | $1.61 \cdot 10^{-3}$ |
| Hierarchy-2 | 29.97 | 0.9570 | 0.0474 | 0.3106 | 1.65 | 0.1137 | $1.74 \cdot 10^{-3}$ |
| Multires-AR-2 | 30.25 | 0.9583 | 0.0454 | 0.2991 | **1.59** | 0.1116 | $1.79 \cdot 10^{-3}$ |
| Reverse AR | 30.04 | 0.9590 | 0.0450 | 0.2920 | 1.8 | 0.1134 | $\mathbf{1.51 \cdot 10^{-3}}$ |

## H.2    QUALITATIVE SAMPLING SCHEME ABLATIONS

In this section we highlight the qualitative impact that different sampling schemes can have on the quality of inpainted videos. We select a video from the test set where conditioning on the appropriate frames is crucial for our method to succeed. Figure 13 shows the beginning of this video, where a car is occluded in the right-hand lane for the first few seconds of the video (see input frames in the first row). The Autoregressive scheme (second row) shows a distinct "pop-in" effect, as when the initial frames were generated the model was not able to condition on future frames where the cars existence and appearance are revealed. Both Reverse-Autoregressive and AR w/ Far Future (third and fourth rows) do condition on future frames that contain the car; Reverse-Autoregressive because the model is able to propagate the car backwards through time, and AR w/ Far Future because the model conditions on frames far out into the future where the car has been revealed. Figure 14 shows

the end of the video, where the cars in the right-hand lane are occluded and remain occluded for the rest of the video (first row). The Autoregressive scheme keeps these cars visible, as it is able to propagate them forward through time (second row). Reverse-Autoreg (third row) fails for the same reason that Autoreg did at the beginning: when the final frames were generated, the model was not able to condition on frames where the vehicles were visible. AR w/ Far Future (fourth row) is again successful; despite the cars not becoming visble again (and thus there is no "future" to condition on), it is able to propagate the cars forward in time as the Autoreg scheme does.

## I  DIFFUSION MODEL SAMPLING DETAILS

As alluded to in the main paper, we train a diffusion model that uses $\epsilon$-prediction to parameterize the score of a distribution over $\mathbf{x}_t$ at each time $t$. This can be used to parameterize a stochastic differential equation (or ordinary differential equation) that morphs samples from a unit Gaussian into approximate samples from the conditional distribution of interest $p_{\text{data}}(\mathbf{x}|\mathbf{y})$ Song et al. (2020). We use the Heun sampler proposed by Karras et al. (2022) to integrate this SDE. Our hyperparameters are $\sigma_{\max} = 1000$, $\sigma_{\min} = 0.002$, $\rho = 7$, $S_{\text{churn}} = 80$, $S_{\max} = \infty$, $S_0 = 0$, and $S_{\text{noise}} = 1$. We use 100 sampling steps (involving 199 network function evaluations) for each experiment except where specified otherwise.

## J  ADDITIONAL QUALITATIVE RESULTS

### J.1  BDD-INPAINTING

Figure 16 and Figure 16 show additional results on tasks from the BDD-Inpainting test set.

### J.2  TRAFFIC-SCENES

Figure 17 and Figure 18 show additional results on tasks from the Traffic-Scenes training set.

### J.3  INPAINTING-CARS

Figure 19 shows additional results on tasks from the Traffic-Scenes training set.

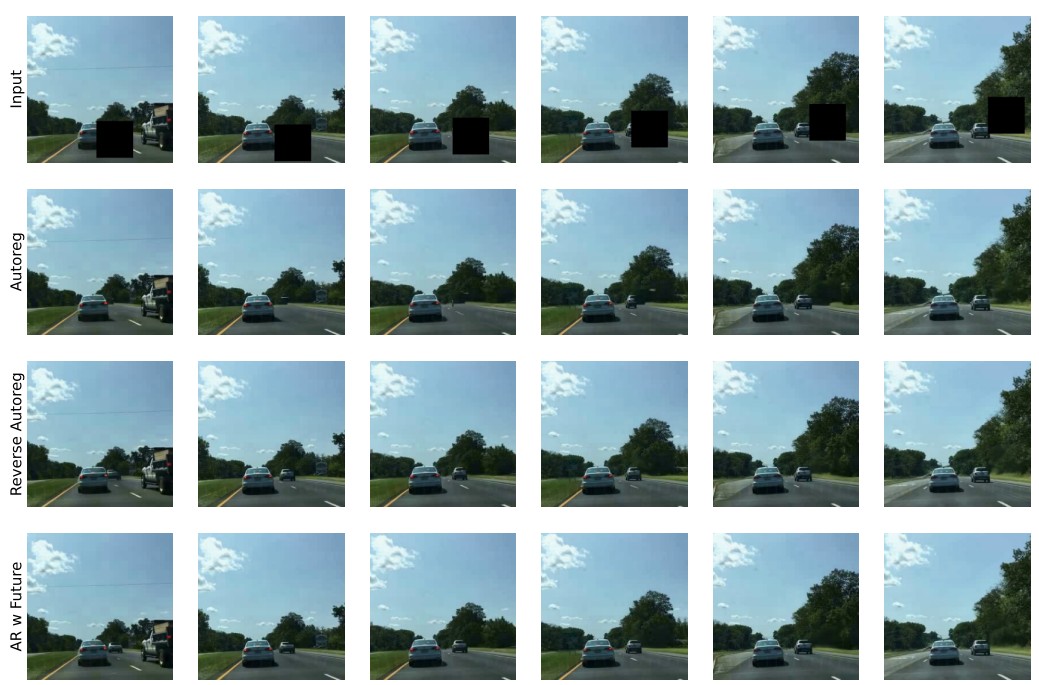

Figure 13: Qualitative results from different sampling schemes on the beginning of the video discussed in Section H.2

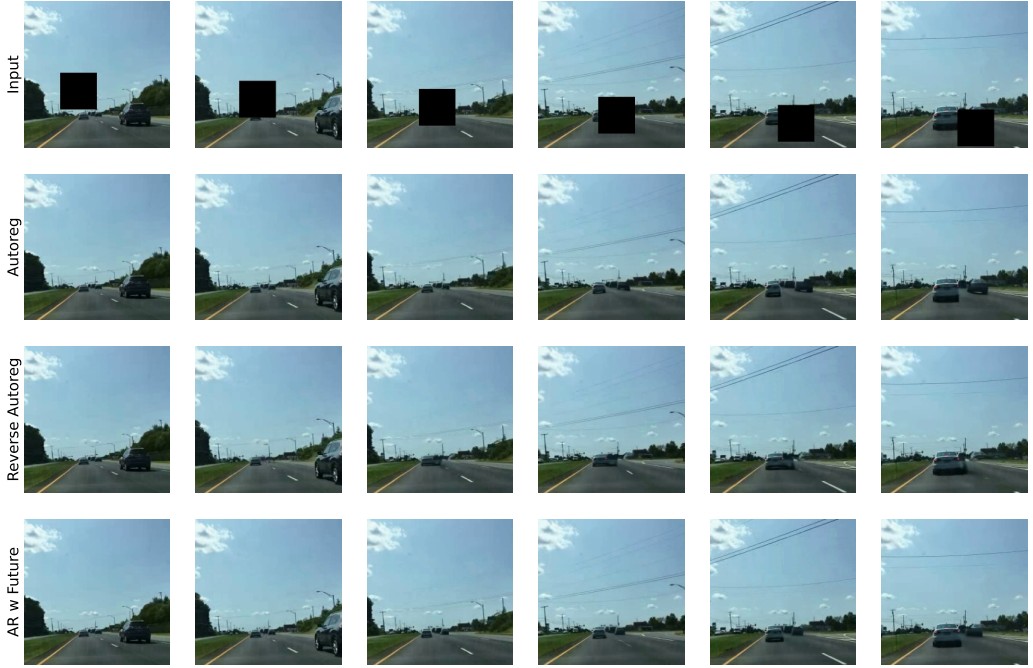

Figure 14: Qualitative results from different sampling schemes on the end of the video discussed in Section H.2

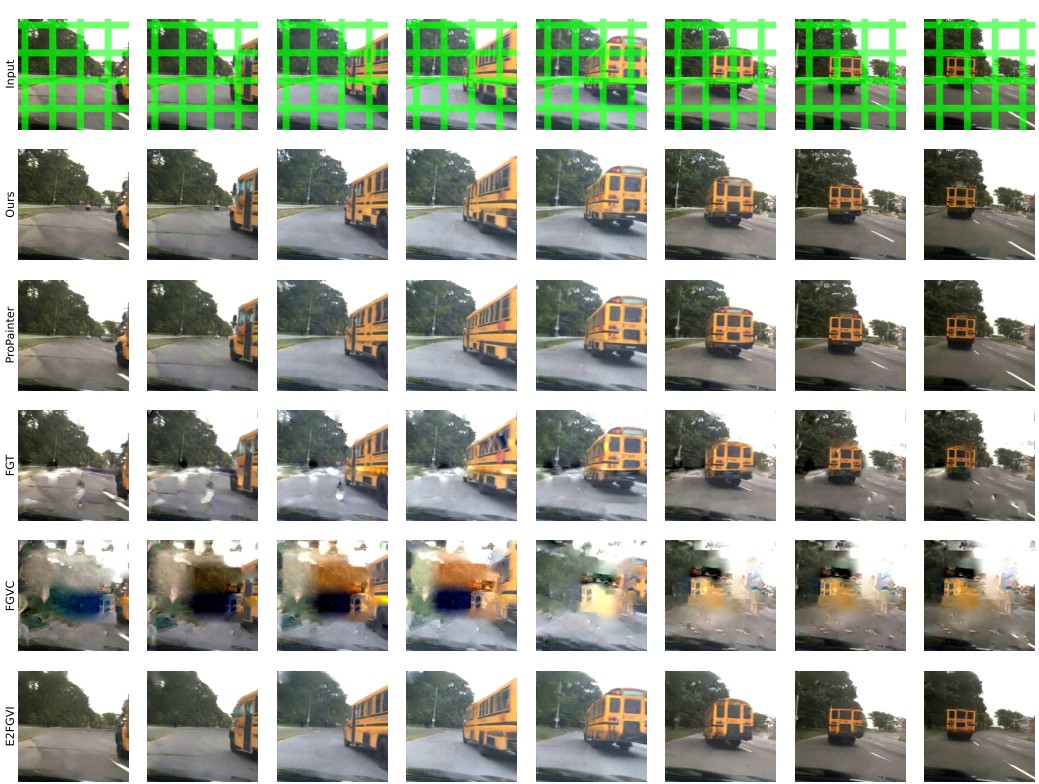

Figure 15: Qualitative results from our method and all the competing methods we compared to quantitatively on an example from the BDD-Inpainting test set. We note that, in the presence of small masks, the qualitative differences are less pronounced between our method and the best-performing benchmarks, like since information in neighbouring frames can more easily be exploited. For our method we use the Lookahead-AR++ sampling scheme.

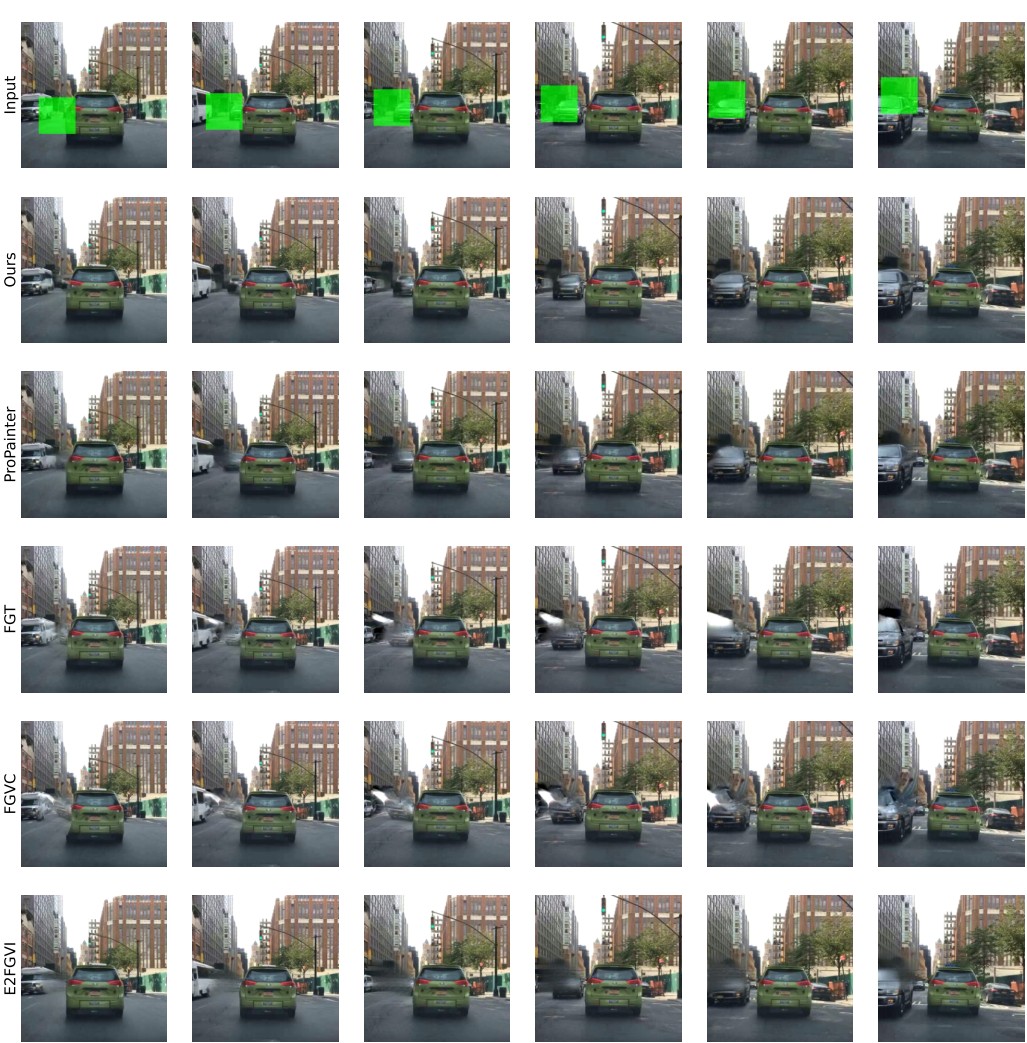

Figure 16: Qualitative results from our method and all the competing methods we compared to quantitatively on an example from the BDD-Inpainting test set. Again, for our method we use the Lookahead-AR++ sampling scheme. In this video the vehicle passing in the left-hand lane is only ever partially visible, causing other methods to produce blurry results.

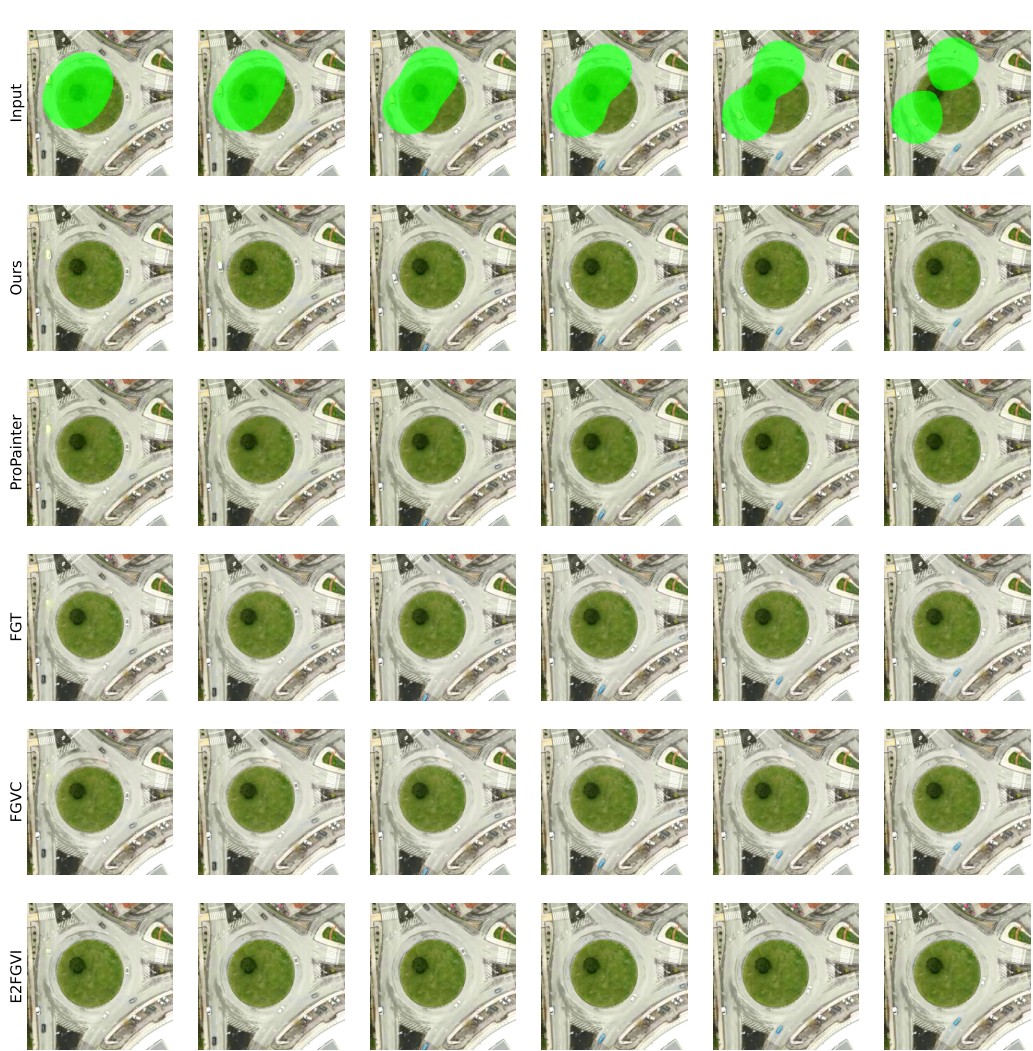

Figure 17: Qualitative results from our method and all the competing methods we compared to quantitatively on an example from the Traffic-Scenes test set. For our method we use the Hierarchy-2 sampling scheme. For our method the two occluded vehicles are inpainted with continuous trajectories; for all other methods the vehicles disappear while they are occluded.

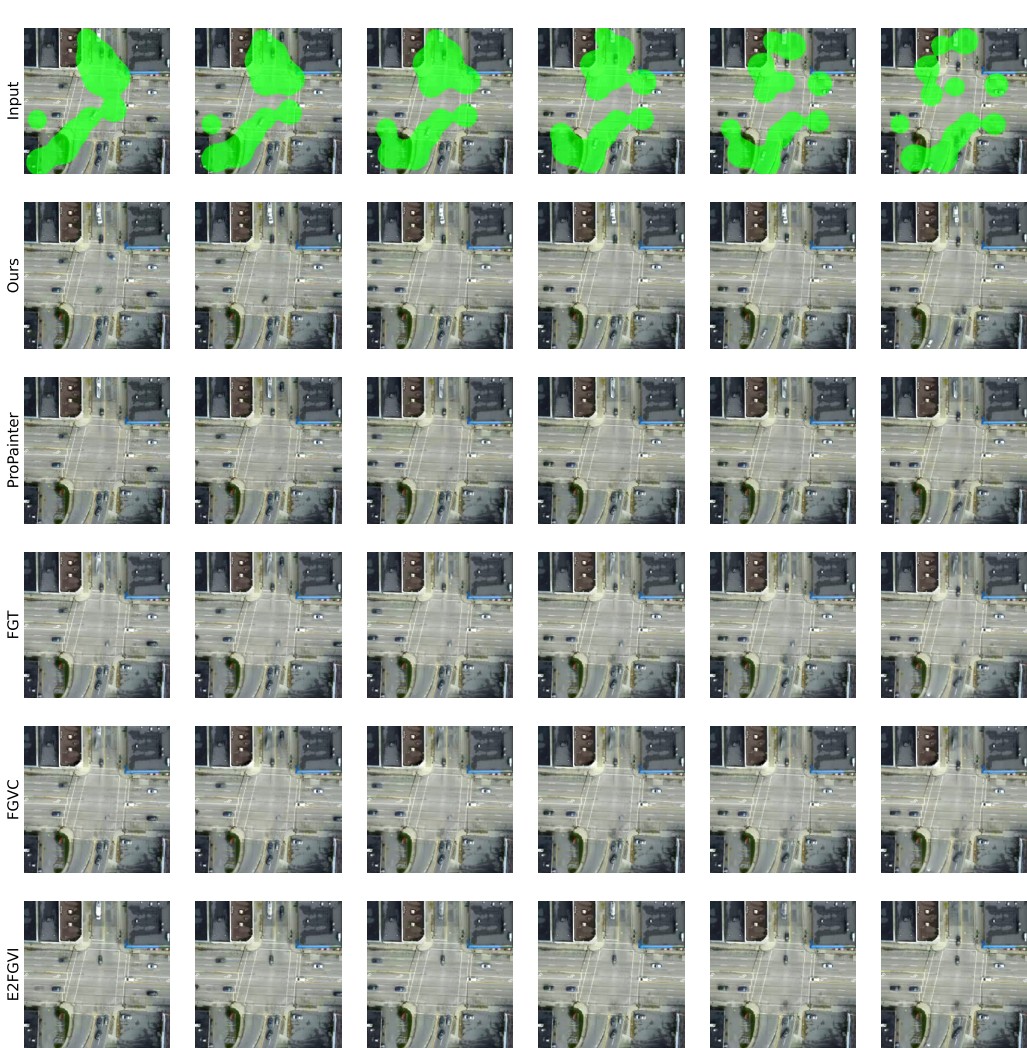

Figure 18: Further qualitative results from our method and all the competing methods we compared to quantitatively on an example from the Traffic-Scenes test set. In the ground truth for this example, in the first frame there is a vehicle making a left hand turn towards the bottom of the image that is occluded until it briefly emerges many frames later. Our model is able to initialize a vehicle in the inpainting and complete a trajectory which is consistent with the exit point for the vehicle. In all other methods the vehicle appears suddenly.

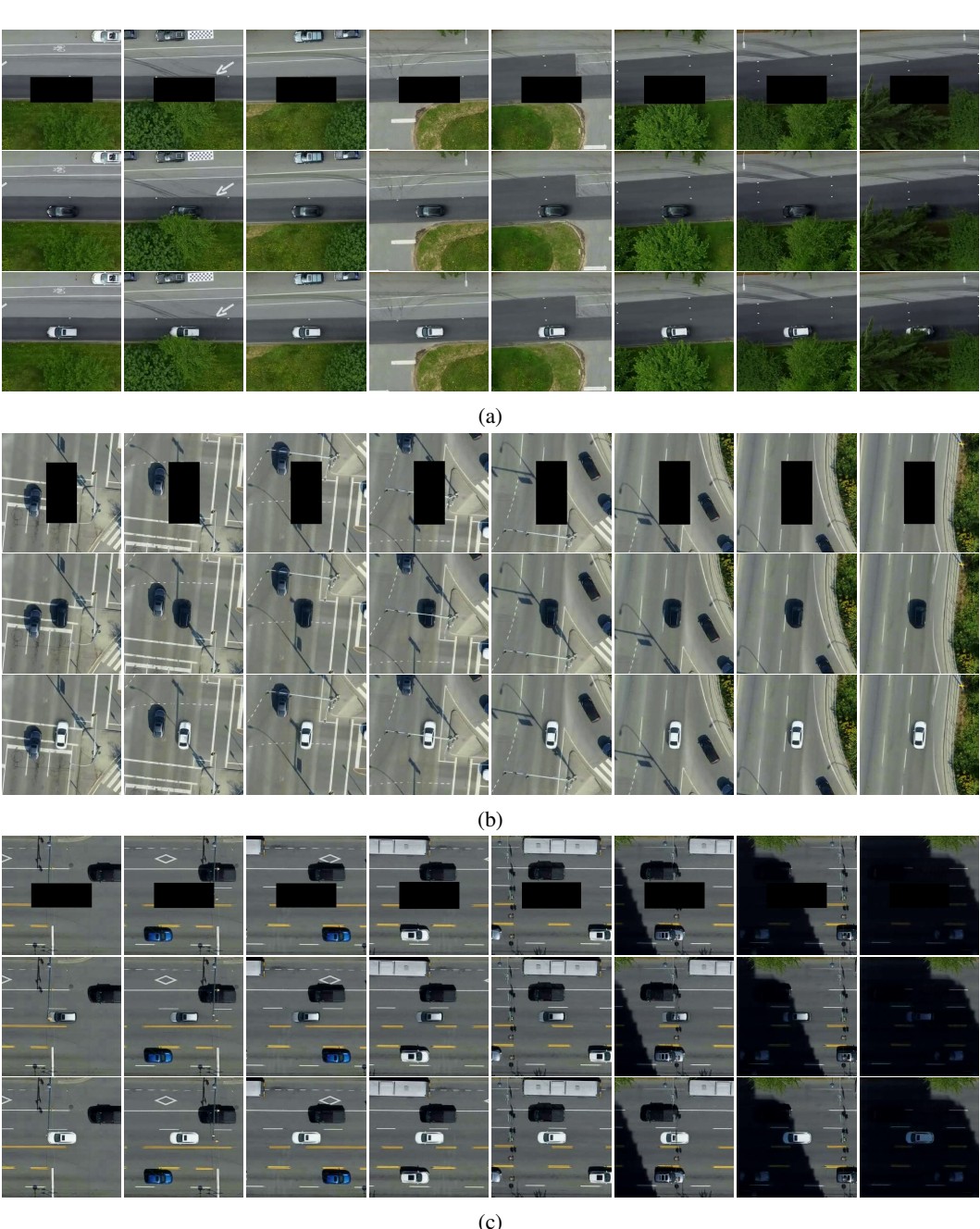

(a)

(b)

(c)

Figure 19: Additional qualitative results from our method on the Inpainting-Cars test set, demonstrating our method's ability to generate diverse vehicle appearances to inpaint the scene and to deal with various complications in the context. In Fig. 19a, our method is able to deal with partial occlusion by trees in the scene. In Fig. 19b, note that the inpainted cars' shadows are oriented in the same direction as the shadows of other objects. Fig. 19c demonstrates that our method is capable of realistically shading an inpainted vehicle's color when it enters the shade.

