# OpenReview forum: "Semantically Consistent Video Inpainting with Conditional Diffusion Models"
_ICLR.cc/2025/Conference — Submitted to ICLR 2025_

### Official Review · Reviewer_i15w · 2024-10-27

**Soundness:** 3
**Presentation:** 3
**Contribution:** 2
**Rating:** 5
**Confidence:** 4

**Summary:**

This paper introduces a novel video inpainting framework based on conditional diffusion models, which can generate semantically consistent content that remains realistic over extended occlusions and diverse scene complexities. The authors present sampling schemes tailored for inpainting that capture essential long-range dependencies within the context and develop a new method to condition on known pixels in incomplete frames. Results show that the proposed method consistently outperforms methods based on optical flow or attention.

**Strengths:**

- The motivation is clear and the techniques sound reasonable.
- The writing and organization are easy to follow.
- The experiment result validates the effectiveness of this approach, especially for complex scenarios where occlusions (input information is scarce) prevent conventional methods from performing well.

**Weaknesses:**

1. The author needs to highlight this paper’s innovative contributions in the INTRODUCTION part and present a completed method graph rather than just showing model inputs.
2. This work relies heavily on the inpainting model to handle object occlusions and interactions. However, the authors do not address how the model performs in highly cluttered or dense environments with multiple possible object placements.
3. Additional experiments are needed to demonstrate the effectiveness of the sampling schemes and the model's generalization to non-traffic datasets, along with further comparisons/discussions with recent works, such as [1] [FFF-VDI](https://arxiv.org/pdf/2408.11402) [2] [FGT++](https://arxiv.org/pdf/2301.10048) [3] [SViT](https://openaccess.thecvf.com/content/ICCV2023/papers/Lee_Semantic-Aware_Dynamic_Parameter_for_Video_Inpainting_Transformer_ICCV_2023_paper.pdf).

**Questions:**

- Could you also provide quantitative comparisons of your method on the YouTube-VOS and DAVIS datasets? Additional results for object removal tasks would be more convincing.

---

> ### Author Response · Authors · 2024-11-22
> **Author Response**
>
> > The author needs to highlight this paper’s innovative contributions in the INTRODUCTION part and present a completed method graph rather than just showing model inputs.
>
> Thank you for this suggestion. We have edited the final paragraph of the introduction to summarize our core contributions, which are as follows:
> - We reframe video inpainting as a conditional generative modeling problem, and present a framework for solving this problem using conditional video diffusion models.
> - We demonstrate how to use long-range temporal attention to generate semantically consistent behaviour of inpainted objects over long time horizons, even when our model cannot jointly model all frames due to memory constraints. We can do this even for inpainted objects that have limited or no visibility in the context, a quality not present in the current literature.
> - We introduce inpainting-specific sampling schemes which capture crucial long-range dependencies in the context, and devise a novel method for conditioning on the known pixels in incomplete frames.
> - We report strong experimental results on several challenging video inpainting tasks, outperforming state-of-the-art approaches on a range of standard metrics.
> - We release the BDD-Inpainting dataset, a large scale video inpainting dataset with two test sets representing a challenging new benchmark for video inpainting methods.
>
> Also, as noted in our response to reviewer Uk6y, we have prepared a diagram outlining the sampling process at a high level which we will include in our revised manuscript, and will also provide detailed training and sampling algorithms in the appendix to remove any remaining ambiguity.
>
> > This work relies heavily on the inpainting model to handle object occlusions and interactions. However, the authors do not address how the model performs in highly cluttered or dense environments with multiple possible object placements.
>
> We believe that both BDD-Inpainting and, in particular, Traffic-Scenes are representative of dense, cluttered inpainting problems, and contain examples which permit different object trajectories which are consistent with the context. Such examples are already present in our current draft in Figures 15-17, and in particular the BDD and Traffic-Scenes example videos in the supplementary materials. If these examples are not satisfactory, we kindly ask the reviewer to further specify what such a dense/cluttered scene might look like.
>
> > Additional experiments are needed to demonstrate the effectiveness of the sampling schemes and the model's generalization to non-traffic datasets, along with further comparisons/discussions with recent works, such as [1] FFF-VDI [2] FGT++ [3] SViT.
>
> Thank you for bringing these relevant works to our attention. Unfortunately, to the best of our knowledge, code is not has not been released for any of them and so a quantitative comparison is not possible. We will, however, discuss them in the related work section of our revised manuscript.
>
> > Could you also provide quantitative comparisons of your method on the YouTube-VOS and DAVIS datasets? Additional results for object removal tasks would be more convincing.
>
> While we agree that such a comparison would be informative, the size of the YouTube-VOS dataset precludes comparison in our training setup. Please also see the general response to all reviewers above.

---

### Official Review · Reviewer_ca5i · 2024-10-28

**Soundness:** 3
**Presentation:** 2
**Contribution:** 2
**Rating:** 5
**Confidence:** 5

**Summary:**

State-of-the-art video inpainting methods focus on propagating visual information but struggle with generating novel content. This paper redefines video inpainting as a conditional generative task using video diffusion models, enabling synthesis of inpaintings with long-range temporal consistency. The approach leverages inpainting-specific sampling schemes to handle incomplete frames and is effective even when objects are partially visible or absent in the initial context.

**Strengths:**

1. The authors invest significant effort in building a large-scale video dataset to train the diffusion model.
2. Compared to previous video inpainting methods, the proposed model emphasizes filling in unseen content rather than relying solely on propagation.

**Weaknesses:**

1. Practicality concerns: The proposed dataset is heavily focused on scenarios involving cars, which restricts its generalizability to broader real-world applications.

2. Lack of control over inpainting content: The proposed model does not offer mechanisms to specify what should fill the missing regions.

3. Limited diversity in demonstrations: All examples shown in the paper involve cars in the inpainted regions. Are there demonstrations with objects other than cars?

4. Comparison with recent methods: Some recent mask-guided video editing techniques can also fill gaps with cars, offering options to specify the car model and color via text prompts. Why is there no comparison with these methods?

5. The claim of “strong experimental results” is potentially biased. The authors only test their model on BDD-Inpainting and Traffic-Scenes (datasets used for model training), while baselines are only trained on Youtube-VOS and DAVIS. This discrepancy in training data limits fair comparison.

**Questions:**

Inference time: How long does the proposed model take to process a typical 2-second video clip?

---

> ### Author Response · Authors · 2024-11-16
> **Request for citation**
>
> We'd like to thank reviewer Ca5i for their review of our work and are working on a detailed response. So that we may better address your feedback, we kindly request a citation for the masked video editing method you are referring to in the fourth point of your weaknesses section.

---

> ### Author Response · Authors · 2024-11-22
> **Author Response**
>
> Thank you for your review. We address the issues you highlighted below.
>
> > Practicality concerns: The proposed dataset is heavily focused on scenarios involving cars, which restricts its generalizability to broader real-world applications.
>
> As our focus in this work was generative inpainting in settings with limited computation, we chose to specialize our experiments to driving scenarios, a domain with clear practical applications. As noted in the response to all reviewers above, video generative models trained to operate on open domains are typically trained on datasets with millions of examples and are prohibitively resource intensive to train. By introducing large scale datasets on restricted domains, we intended to enable ourselves and hopefully others to pursue research on generative approaches to video inpainting with a limited compute budget. Further, these datasets were designed to highlight the shortcomings of the current state-of-the-art approaches to video inpainting (namely their inability to synthesize novel content not present in the context), and spur further research aimed at addressing these shortcomings.
>
> > Lack of control over inpainting content: The proposed model does not offer mechanisms to specify what should fill the missing regions.
>
> In this work we chose to focus on unconditional inpainting, as do all of the works which we compare to. While some recent works such as [AVID](https://arxiv.org/abs/2312.03816) and [CoCoCo](https://arxiv.org/abs/2403.12035) focus on text-conditional video inpainting, we believe this line of work is quite distinct from the majority of the (unconditional) video inpainting literature in terms of both the nature of the task and the potential downstream applications. Whereas text-conditional inpainting models must integrate the context with the user-specified content, an unconditional model has a harder task as it must decide for itself which objects to inpaint. Further, while text-conditional approaches tend to emphasize their ability to edit the appearance of objects in videos according to user preferences, the focus in unconditional inpainting tends to be on *e.g.* occlusion removal without the need for a human-in-the-loop. That being said, we believe extending our framework to allow for prompt-based guidance in a manner similar to AVID or CoCoCo would be an exciting future research opportunity.
>
> > Limited diversity in demonstrations: All examples shown in the paper involve cars in the inpainted regions. Are there demonstrations with objects other than cars?
>
> As our models are trained to model the conditional data distribution defined by our objective function, it is true that our method will only produce inpaintings with content similar to what is observed in the training set. However, while all of our datasets are thematically similar in the sense that they involve cars, we believe they represent a diverse set of inpainting problems in terms of what the model is required to learn to perform well. For instance, BDD-Inpainting requires the model to deal with occluded objects changing their position relative to the camera, while Traffic-Scenes requires the model to infer semantically reasonable trajectories for vehicles over long time horizons. We also note that these limitations are not inherent to the framework we have proposed – our method does not incorporate any inductive biases tailored specifically to vehicle motion which would reduce its generalizability to other real-world applications given an appropriate training set.
>
> > Comparison with recent methods: Some recent mask-guided video editing techniques can also fill gaps with cars, offering options to specify the car model and color via text prompts. Why is there no comparison with these methods?
>
> As mentioned in our response to your second point, we believe text-conditional inpainting to be a distinct task from the unconditional inpainting task we focus on here. While we did cite [AVID](https://arxiv.org/abs/2312.03816), we did not consider comparing to it for two main reasons: (a) for most examples in our datasets the appearance of occluded object is at least partially specified by the context, and so prompting itself becomes difficult as the prompt must be consistent with the context; (b) it is unclear to us how one would perform an unbiased quantitative comparison between conditional and unconditional models as the prompts used would have a significant impact on the results.

---

> ### Author Response · Authors · 2024-11-22
> **Author Response (continued)**
>
> > The claim of “strong experimental results” is potentially biased. The authors only test their model on BDD-Inpainting and Traffic-Scenes (datasets used for model training), while baselines are only trained on Youtube-VOS and DAVIS. This discrepancy in training data limits fair comparison.
>
> While it is true that the methods we compare to were only trained on YouTube-VOS, we note that it is common in the video inpainting literature to assess such methods on datasets that they were not trained on. Of particular note is the fact that DAVIS is only used as a test set with no corresponding training set, and nevertheless is one of the standard benchmarks used in the video inpainting literature, and the recent work [DEVIL](https://arxiv.org/abs/2105.05332) proposes another benchmark test set with no corresponding training set. Further, these works tend to emphasize their ability to function as general purpose inpainting methods, prominently featuring inpainting results produced by their models on movies or other out-of-distribution inputs on their websites and demo videos (see [ProPainter](https://shangchenzhou.com/projects/ProPainter/) and [E2FGVI](https://www.youtube.com/watch?v=N--qC3T2wc4)). The qualitative results of these methods on our datasets are extremely impressive (see e.g. Figures 14 and 15 in our paper), and they tend to perform similarly to our method on all examples except those which explicitly require novel content generation. As such we believe that these pre-trained models serve to quantify a baseline level of competency in video inpainting on our datasets for our method to compare against, which is a testament to these models’ generalizability on tasks where content propagation is sufficient to produce high quality results.
>
> > Inference time: How long does the proposed model take to process a typical 2-second video clip?
>
> While the clips we evaluated on were ten seconds long rather than two seconds, we provide a comparison of runtimes between our method and those we compare to in our response to reviewer Uk6y.

---

> > ### Comment · Reviewer_ca5i · 2024-11-25
> >
> > I don’t think it is common practice to test on an untrained dataset. If it is, then why not train the proposed method on the introduced BDD100K dataset and evaluate it on YouTube-VOS and DAVIS? Instead, the baselines are trained on YouTube-VOS and DAVIS and then tested on your proposed BDD100K dataset.
> >
> > Additionally, since the rebuttal states that this work focuses on unconditional inpainting, why wasn’t this explicitly clarified in the main text? This omission can easily lead to confusion, as readers would naturally expect comparisons with recent text/mask-guided video inpainting methods.
> >
> > Overall, I maintain my score as I recognize the effort in creating the dataset. However, for other aspects, particularly experimental comparisons, it is difficult to fairly verify the novelty of the method.

---

> ### Author Response · Authors · 2024-11-26
>
> Thank you for your response.
>
> **Regarding testing on untrained datasets**
>
> It is indeed common practice to train exclusively on YouTube-VOS and test on both YouTube-VOS and DAVIS. We quote the relevant text from methods we compare to below.
>
> [ProPainter](https://arxiv.org/abs/2309.03897):
> > We use the training set of YouTube-VOS with
> 3471 video sequences to train our networks. Two widely-
> used test sets are adopted for evaluation: YouTube-VOS
> and DAVIS, which consist of 508 and 90 video se-
> quences, respectively.
>
> [FGT](https://arxiv.org/abs/2208.06768):
> > We adopt Youtube-VOS and DAVIS datasets for evaluation. Youtube-
> VOS contains 4453 videos and DAVIS contains 150 videos. We adopt the training
> set of Youtube-VOS to train our networks. As for Youtube-VOS, we evaluate the
> trained models on its testset. Since DAVIS contains densely annotated masks on
> its training set, we adopt its training set to evaluate our method.
>
> [E2FGVI](https://arxiv.org/abs/2204.02663):
> > We train our model on the YouTube-VOS dataset and evaluate it on both
> YouTube-VOS and DAVIS datasets.
>
> Similar passages can be found in [Fuseformer](https://arxiv.org/abs/2109.02974), [ISVI](https://openaccess.thecvf.com/content/CVPR2022/papers/Zhang_Inertia-Guided_Flow_Completion_and_Style_Fusion_for_Video_Inpainting_CVPR_2022_paper.pdf), [FFF-DVI](https://arxiv.org/abs/2408.11402), [FGT++](https://arxiv.org/abs/2301.10048), [SViT](https://openaccess.thecvf.com/content/ICCV2023/papers/Lee_Semantic-Aware_Dynamic_Parameter_for_Video_Inpainting_Transformer_ICCV_2023_paper.pdf), etc. Note that FGVC is somewhat non-standard in its experimental setup, as it evaluates only on DAVIS and trains only on a 60 video subset of DAVIS, which it is able to do as it makes heavy use of pretrained modules and only a flow-edge completion network is trained.
>
> We believe the impressive generalization capabilities of these approaches lie in the fact that they learn to complete optical flow fields, and that this procedure is largely agnostic to the semantic content in the videos. However, as noted in our paper, this approach falls short when a realistic inpainting requires new content to be generated. As our generative approach learns a conditional distribution over inpaintings given the observed content, it takes the semantics of the video into account. As a result our method is more specialized to the content of the dataset it was trained on, but gains the ability to inpaint unseen content into videos in a semantically consistent manner.
>
> **Regarding conditional vs. unconditional inpainting**
>
> We believe our paper is sufficiently clear about the problem it is solving, and it makes mention of the alternative task of text-conditional video inpainting in the related work section. As the vast majority of the video inpainting literature focuses on "unconditional" inpainting, we do not believe that explicitly using this terminology is necessary.

---

### Official Review · Reviewer_jGsq · 2024-10-28

**Soundness:** 4
**Presentation:** 3
**Contribution:** 3
**Rating:** 6
**Confidence:** 3

**Summary:**

The authors extend the idea of Flexible Diffusion Model (FDM) to the video inpainting domain. A different sampling and training scheme is also developed to accomodate the new task.

**Strengths:**

- The paper is easy to follow
- The idea is intuitive and works also practically well
- The authors showcased that their model is better than flow- and attention-based video inpainting methods both qualitatively and quantitatively
- Different sampling schemes are also evaluated extensively.

**Weaknesses:**

Given that the authors proposed a variety of inference techniques, it would be interesting to investigate whether different sampling schemes during **training** makes a difference. It seems from Equation 3 that the $\mathcal{X}$ and $\mathcal{Y}$ are randomly sampled; would a benefit be gained e.g., when training the model in a similar way to how you sample at inference?

**Questions:**

Is the model trained from scratch? How well does it generalize to other, more complicated open video domains?

---

> ### Author Response · Authors · 2024-11-22
> **Author Response**
>
> > Given that the authors proposed a variety of inference techniques, it would be interesting to investigate whether different sampling schemes during training makes a difference. It seems from Equation 3 that the $\mathcal{X}$ and $\mathcal{Y}$ are randomly sampled; would a benefit be gained e.g., when training the model in a similar way to how you sample at inference?
>
> We considered this early on and found that there was some benefit to doing this for simple sampling schemes like Autoregressive, where the relative positions of frames indices $\mathcal{X}$ and $\mathcal{Y}$ are the same for all videos and hence the training distribution can be made to exactly match the test-time distribution. For more complicated schemes like Lookahead-AR++ however, the relative positions of $\mathcal{X}$ and $\mathcal{Y}$ depend not only on the stage $s$, but also on the length $N$ of the video being inpainted (see e.g. Figure 5(b) vs. Figure 8(g)). As such, in order to match the test-time distribution over $\mathcal{X}$ and $\mathcal{Y}$, you would still need to randomize over $N$ for all video lengths which you would like to hypothetically be able to inpaint. We decided instead to train the model to be able to make predictions for any $\mathcal{X}$ given any $\mathcal{Y}$ as this had been shown to work well in practice and had the added benefit of allowing for test-time experimentation with sampling schemes, which offers practical benefits as any predetermined sampling scheme may not capture the important conditional dependencies for a given input.
>
> > Is the model trained from scratch? How well does it generalize to other, more complicated open video domains?
>
> We train a separate model from scratch for each of the datasets on which we evaluate. Since our training objective trains the model to reproduce the conditional data distribution of a specific data distribution, we would not expect these models to naturally generalize to inputs which differ significantly to those in our datasets.

---

> > ### Comment · Reviewer_jGsq · 2024-11-26
> >
> > Thank you for the effort you put into the rebuttal, it effectively addresses my concerns. While I continue to believe that this paper presents a strong and innovative idea which appears naturally extendable to other datasets, I note that the other reviewers have consistently raised the issue of the lack of evaluation on widely-used benchmarks, such as YouTube-VOS and DAVIS. This limitation remains unaddressed, and no empirical evidence has been provided in the rebuttal to counter this point.
> >
> > In light of these considerations, I will adjust my score accordingly.

---

### Official Review · Reviewer_Uk6y · 2024-11-04

**Soundness:** 3
**Presentation:** 3
**Contribution:** 2
**Rating:** 6
**Confidence:** 2

**Summary:**

This paper presents a conditional generative framework for video inpainting that leverages diffusion models to synthesize new, semantically coherent content for large occlusions. Unlike traditional methods dependent on optical flow, this approach achieves realistic object behavior and maintains temporal consistency even when entire content regions must be generated. Evaluated on newly constructed datasets, the framework demonstrates superior quality and consistency in inpainted sequences compared to existing methods.

**Strengths:**

1. The method effectively captures temporal and spatial dependencies, enabling high-quality synthesis for occluded scenes and objects.
2. Quantitative results show clear improvements over state-of-the-art methods, especially in challenging video inpainting scenarios.

**Weaknesses:**

1. Please provide inference costs (e.g., time and peak GPU memory usage) for a specified input size and number of frames, and compare these with other SOTA methods.

2. A dedicated ablation study comparing the method trained on standard datasets used by other methods versus the newly introduced datasets is suggested. This would help isolate the effects of the method itself from those of the datasets, as data alone can often address significant aspects of the problem.

3. The framework lacks clear illustration, making it difficult to grasp the main concepts. For instance, it is unclear how conditional inputs are processed in the forward pass, the roles of the trainable and frozen modules, or if any new layers for controllable generation are introduced.

4. Please also discuss the potential limitations or failure cases of the proposed method. Beyond inpainting cars, can this method inpaint other objects in a zero-shot manner?

**Questions:**

Please see the weaknesses.

---

> ### Author Response · Authors · 2024-11-22
> **Author Response**
>
> Thank you for your review. We address the issues you highlighted below.
>
> > Please provide inference costs (e.g., time and peak GPU memory usage) for a specified input size and number of frames, and compare these with other SOTA methods.
>
> We have compiled this information below for two separate 100 frame subsequences from the BDD dataset using an NVIDIA GeForce RTX 2080 Ti.
>
> **Example from Figure 1**
> |                  | Runtime | Peak GPU Memory (GB) |
> |------------------|---------|----------------------|
> | ProPainter       | 1m7s    |         2.68         |
> | E2FGVI           | 0m54s   |         3.47         |
> | FGVC (seamless)  | 228m43s |         0.13         |
> | FGVC             | 18m48s  |         0.13         |
> | FGT              | 6m27s   |         4.23         |
> | Ours (100 steps) | 9m22s   |         7.29         |
> | Ours (10 steps)  | 1m6s    |         7.29         |
>
> **Example from Figure 14**
> |                  | Runtime | Peak GPU Memory (GB) |
> |------------------|---------|----------------------|
> | ProPainter       | 0m32s   |         2.84         |
> | E2FGVI           | 0m19s   |         3.47         |
> | FGVC (seamless)  | 15m27s  |         0.13         |
> | FGVC             | 4m36s   |         0.13         |
> | FGT              | 11m19s  |         4.23         |
> | Ours (100 steps) | 9m58s   |         7.29         |
> | Ours (10 steps)  | 1m6s    |         7.29         |
>
> We decided to include two examples as the inference costs for the methods that we compare against depend not only on the number of frames but also on the mask and content of the videos being inpainted, and can vary significantly between videos. For FGVC we include results both with and without the “seamless” flag, which the authors claim produces superior results at the cost of increased processing time. Note that the results for FGVC reported in the paper are computed using the “seamless” option. Additionally, we include inference costs for our method using both 100 and 10 Heun sampler steps, recalling that our method allows for a tradeoff between computational cost and inpainting quality. While our method has the highest memory requirement out of those we compare to, it remains competitive in terms of runtime, particularly using fewer sampling steps.
>
> > A dedicated ablation study comparing the method trained on standard datasets used by other methods versus the newly introduced datasets is suggested. This would help isolate the effects of the method itself from those of the datasets, as data alone can often address significant aspects of the problem.
>
> While we agree that such a comparison would be informative, we did not train a model on the YouTube-VOS dataset for the reasons noted in the response to all reviewers above.
>
> > The framework lacks clear illustration, making it difficult to grasp the main concepts. For instance, it is unclear how conditional inputs are processed in the forward pass, the roles of the trainable and frozen modules, or if any new layers for controllable generation are introduced.
>
> Thank you for bringing this point to our attention. We have prepared a diagram outlining the sampling process at a high level which we will include in our revised manuscript, and will also provide detailed training and sampling algorithms in the appendix to remove any remaining ambiguity. In the meantime we will address your questions briefly here:
> - At each denoising step in the generative process, for the selected frames both the noised and ground truth pixels along with the masks are provided to the model as input, as illustrated in Figure 4. The model outputs predictions for the denoised values of all pixels in the input, but only the values of the unknown pixels (as indicated by the masks) are updated, such that the known pixels retain their ground truth values at each step.
> - Our method is trained in a fully end-to-end manner and does not involve any frozen modules.
> - We do not address controllable generation in this work and, as such, no such layers are included in our models.

---

> > ### Comment · Reviewer_Uk6y · 2024-11-26
> >
> > I share the reviewers' concerns regarding the fairness of comparisons with other SOTA methods, given that different training datasets were used for various baselines. To properly evaluate the contributions of the collected dataset or the proposed method itself, it is crucial to either re-train the proposed method on prior datasets or re-train the baselines using the proposed dataset. Additionally, the method’s focus on traffic-related scenes may significantly limit its applicability to inpainting or generating other natural objects. That said, I do recognize and appreciate the authors' effort in creating this dataset, which is a valuable contribution in its own right. Thus, I will maintain my initial rate for this paper.

---

> ### Author Response · Authors · 2024-11-22
> **Author Response (continued)**
>
> > Please also discuss the potential limitations or failure cases of the proposed method. Beyond inpainting cars, can this method inpaint other objects in a zero-shot manner?
> We briefly discuss the potential limitations of our method in Appendix A. From our perspective, the most notable limitations of our method are:
> - Computation cost relative to some competing methods, as further detailed above.
> - Our method’s need to be trained on a large dataset which is semantically similar to the data one wishes to inpaint at test time.
>
> Regarding the ability to inpaint other objects in a zero-shot manner: since we frame video inpainting as a conditional generative modeling problem, our method produces results such that the content of the inpainted regions matches the conditional distribution of the training data. While framing the problem in this manner can produce diversity in the inpainted regions, we are fundamentally limited by the distribution of the training data. As such, we would not expect our method to be able to inpaint object classes which are not present in the training set. We note that this is not a limitation of our framework but rather of the models that we have trained to date.

---

### Author Response · Authors · 2024-11-22
**Response to all reviewers**

We thank all of the reviewers for their thoughtful comments and valuable feedback. We will soon post a revised manuscript addressing many of the points raised here, and believe our paper will be much stronger as a result. In particular, the following changes will be made:

- Adding run time and peak GPU memory usage for our method and those we compare to, as requested by reviewer Uk6y.
- Adding a diagram illustrating our method’s inpainting process, as requested by reviewers Uk6y and i15w, along with detailed training and sampling algorithms to remove any ambiguity about our approach.
- An updated introduction which itemizes the core contributions of our work, as suggested by reviewer i15w.

If we missed anything that reviewers feel should be changed in the paper, or if any questions are not addressed satisfactorily, please send us a reply and we will be happy to make further changes or expand upon our answers.

## Regarding the standard YouTube-VOS and DAVIS benchmarks

Three out of four reviewers asked about the omission of these data sets from our paper, and so we felt it best to address this in a general comment and address the finer points of these questions in our responses to individual reviewers.

As video generative models attempt to model the distribution from which their training data was generated, they require large scale datasets which provide reasonable coverage of this distribution in order to produce visually compelling results. We note that both the number of videos in the dataset as well as the number of frames per video contribute to the meaning of “large scale” here, as such models typically subsample frames due to memory constraints. Further, as the content of the videos becomes more semantically diverse, we expect that more training examples are needed in order to cover the support of the distribution. Even on restricted and relatively simple domains, common training datasets for video generation such as [UCF-101](https://arxiv.org/abs/1212.0402), [BAIR Robot Pushing](https://arxiv.org/abs/1710.05268) and [Kinetics-600](https://arxiv.org/abs/1808.01340v1) contain at minimum tens of thousands of examples, while datasets used to train models on open domains such as [WebVid-10M](https://arxiv.org/abs/2104.00650) and [HD-VILA-10M](https://arxiv.org/abs/2111.10337) typically contain millions of videos and require extensive computational resources to train on. As such, the YouTube-VOS training set (note that DAVIS has no training set and is used only as a test set for models trained on YouTube-VOS) is problematic for training generative models, as the videos are semantically diverse and the scale of the dataset is small relative to datasets typically used to train video generative models (3471 videos containing approximately 200 frames each). As a result, we made the decision early on to curate our own large scale datasets on restricted domains, which we hope will enable more research groups working in resource constrained settings to be able to pursue research making use of video generative models in the video inpainting context.

---

### Author Response · Authors · 2024-11-28
**Revised Manuscript Posted**

We thank all of the reviewers for engaging in the discussion period. This is just a quick note to announce that we have now posted the revised manuscript with the previously mentioned changes.

---

### Meta-Review · Area_Chair_cVMT · 2024-12-20

**Metareview:**

Summary: This paper introduces a conditional diffusion model for video inpainting over a long temporal context. Unlike traditional approaches, it avoids reliance on optical flow and presents an inpainting-specific sampling scheme.
Strengths: The proposed approach is intuitive and clearly motivated. It demonstrates advantages over flow- and attention-based methods, as shown through both qualitative and quantitative evaluations.
Weaknesses: As noted by three reviewers, the paper lacks key experimental results and ablations. In particular, there is no ablation study comparing performance on standard datasets versus the newly introduced ones, which would help isolate the method’s contributions from the dataset’s impact. Furthermore, the methodology’s scope is primarily focused on traffic-related scenes, limiting its broader applicability. Reviewer i15w also highlighted the absence of ablations on the sampling scheme's generalization to non-traffic datasets, an issue that remains unaddressed by the authors. The rebuttal has not sufficiently alleviated these concerns to justify overriding the reviewers’ assessments.

**Additional Comments On Reviewer Discussion:**

The paper has several areas that require further refinement. Despite the rebuttal, reviewers maintained their initial assessments, resulting in scores evenly split between marginally below and marginally above the acceptance threshold. While the paper offers interesting contributions, particularly from a dataset perspective, it does not currently meet the acceptance threshold for ICLR.
I encourage the authors to address the reviewers’ feedback by including the remaining experiments and ablations. Revising the methodology to address its limitations and repositioning the paper accordingly would strengthen its case for submission to future venues.

---

### Decision · Program_Chairs · 2025-01-22

Reject